# Finding Neurons in a Haystack:
# Case Studies with Sparse Probing

**Wes Gurnee**                                                                  *wesg@mit.edu*
*Massachusetts Institute of Technology*

**Neel Nanda**                                                            *neelnanda27@gmail.com*
*Independent*

**Matthew Pauly**                                                     *mpauly@college.harvard.edu*
*Harvard University*

**Katherine Harvey**                                                  *kharvey@college.harvard.edu*
*Harvard University*

**Dmitrii Troitskii**                                               *troitskii.d@northeastern.edu*
*Northeastern University*

**Dimitris Bertsimas**                                                         *dbertsim@mit.edu*
*Massachusetts Institute of Technology*

**Reviewed on OpenReview:** *https://openreview.net/forum?id=JYs1R9IMJr*

## Abstract

Despite rapid adoption and deployment of large language models (LLMs), the internal computations of these models remain opaque and poorly understood. In this work, we seek to understand how high-level human-interpretable features are represented within the internal neuron activations of LLMs. We train $k$-sparse linear classifiers (probes) on these internal activations to predict the presence of features in the input; by varying the value of $k$ we study the sparsity of learned representations and how this varies with model scale. With $k = 1$, we localize individual neurons that are highly relevant for a particular feature and perform a number of case studies to illustrate general properties of LLMs. In particular, we show that early layers make use of sparse combinations of neurons to represent many features in superposition, that middle layers have seemingly dedicated neurons to represent higher-level contextual features, and that increasing scale causes representational sparsity to increase on average, but there are multiple types of scaling dynamics. In all, we probe for over 100 unique features comprising 10 different categories in 7 different models spanning 70 million to 6.9 billion parameters.

## 1 Introduction

Neural networks are often conceptualized as being flexible "feature extractors" that learn to iteratively develop and refine suitable representations from raw inputs (Bengio et al., 2013; Donahue et al., 2014). This begs the question: what features are being represented, and how? Probing is a standard technique used to study if and where a neural network represents a specific feature by training a simple classifier (a probe) on the internal activations of a model to predict a property of the input (Belinkov, 2022) (e.g., classifying the tense of a verb based on the activations of a specific layer). Because models are parameterized as a series of dense matrix multiplications and elementwise nonlinearities, a natural intuition is that features are represented as linear directions in activation space (Elhage et al., 2022b) and are iteratively combined to synthesize increasingly

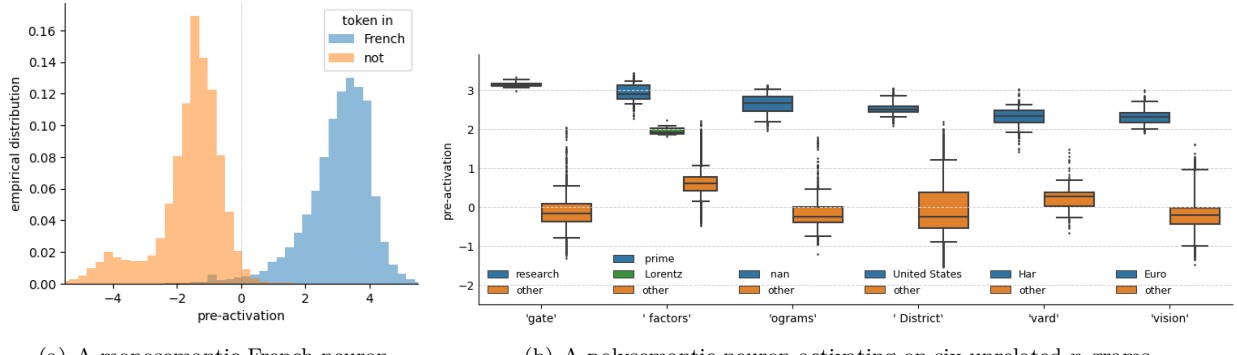

(a) A monosemantic French neuron

(b) A polysemantic neuron activating on six unrelated $n$-grams

Figure 1: Neurons that respond to single features (left) can be understood independently, in contrast to polysemantic neurons (right) that activate for many unrelated features (in this case, the occurrence of specific multi-token spans).

abstract features using linear combinations of previously computed features (Olah et al., 2020b). To zoom in on these dynamics in modern transformer language models, we propose *sparse probing*, where we constrain the probing classifier to use at most $k$ neurons in its prediction; we probe for over 100 features to precisely localize relevant neurons and elucidate broader principles of how to study and interpret the rich structure within LLMs.

In the simplest case, one might hope there exists a 1:1 correspondence between features of the input and neurons in a network—that for the correct feature definition, a 1-sparse probe would be sufficient. Although the literature contains many examples of such seemingly *monosemantic neurons* (Radford et al., 2017; Bau et al., 2020; Olah et al., 2020b; Goh et al., 2021; Cammarata et al., 2021; Elhage et al., 2022a), an obvious problem arises when a network has to represent more features than it has neurons. To accomplish this, a model must employ some form of compression to embed $n$ features in $d < n$ dimensions. While this *superposition* of features enables more representational power, it also causes loss-increasing "interference" between non-orthogonal features (Elhage et al., 2022b). Recent works in toy models demonstrate that this tension manifests in a spectrum of representations: dedicated dimensions or neurons for the most prevalent and important features with increasing levels of superposition—and hence decreasing levels of sparsity—for the long tail of rarer or less important features (Elhage et al., 2022b; Scherlis et al., 2022).

As with any conceptual insight, these results raise more questions than they answer: To what extent do these observations transfer to full scale language models? What kinds of features do or do not appear in superposition? At what scale? How do we reliably find and verify such (feature, neuron) pairs? We leverage sparse probing to systematically study such questions, finding clean examples of both monosemantic neurons and superposition "in the wild." Better understanding—and potentially resolving—superposition is on the critical path to ambitious full-model interpretability because a logical consequence of superposition is the presence of *polysemantic* neurons that activate for a large collection of seemingly unrelated stimuli (Figure 1) (Elhage et al., 2022a; Mu & Andreas, 2020). Such tangled representations undermine our ability to decompose networks into independently meaningful and composable components, thwarting existing approaches to reverse-engineer networks (Olah et al., 2020b).

In addition to being especially well suited to studying superposition, our sparse probing methodology addresses several shortcomings identified in the probing literature (Belinkov, 2022; Antverg & Belinkov, 2021; Hewitt & Liang, 2019; Ravichander et al., 2020). By leveraging recent advances in optimal sparse prediction (Bertsimas et al., 2020; 2021), we are able to prove optimality of the $k$-sparse feature selection subproblem (for small $k$), addressing the conflation of ranking quality and classification quality raised in (Antverg & Belinkov, 2021). Second, by employing sparsity as an inductive bias, our probes maintain a strong simplicity prior and are more capable of precise localization of important neurons. This precision enables a more granular level of analysis illustrated throughout our case studies. Finally, the lack of capacity inhibits the probe from memorizing correlation patterns associated with the feature of interest (Hewitt & Liang, 2019), offering a

more reliable signal of whether this feature is explicitly represented and used downstream (Ravichander et al., 2020).

In the first part of the paper, we outline several variants of sparse probing, discuss the various subtleties of applying sparse probing, and run a large number of probing experiments. In particular, we probe for over 100 unique features comprising 10 different categories in 7 different models spanning 2 orders of magnitude in parameter count (up to 6.9 billion). The majority of the paper then focuses on zooming in (Olah et al., 2020b) on specific examples of general phenomena in a series of more detailed case studies to demonstrate:

- There is a tremendous amount of interpretable structure within the neurons of LLMs, and sparse probing is an effective methodology to locate such neurons (even in superposition), but requires careful use and follow-up analysis to draw rigorous conclusions.

- Many early layer neurons are in superposition, where features are represented as sparse linear combinations of polysemantic neurons, each of which activates for a large collection of unrelated $n$-grams and local patterns. Moreover, weight statistics combined with insights from toy models suggest that the first 25% of fully connected layers employ substantially more superposition than the rest.

- Higher-level contextual and linguistic features (e.g., `is_python_code`) are seemingly encoded by monosemantic neurons, predominantly in middle layers, though conclusive statements about monosemanticity remain methodologically out of reach.

- As models increase in size, representation sparsity increases on average, but different features obey different dynamics: some features with dedicated neurons emerge with scale, others split into finer grained features with scale, and many remain unchanged or appear somewhat randomly.

## 2 Related Work

**Probing**   Originally introduced by (Alain & Bengio, 2016), probing is a standard technique used to determine whether models represent specific features or concepts (Belinkov, 2022). For language models in particular, there exist many probing studies showing the rich linguistic representations learned by models (Conneau et al., 2018; Tenney et al., 2019a;b), contributing to the broader field of "BERTology" (Rogers et al., 2020). Particularly relevant to our work are investigations on the role of individual neurons (Bau et al., 2020; Suau et al., 2020; Sajjad et al., 2022; Wang et al., 2022b) and the identification of sparsely represented features (Durrani et al., 2020; Dalvi et al., 2019). While probing in general has a number of limitations (Hewitt & Liang, 2019; Donnelly & Roegiest, 2019; Ravichander et al., 2020; Maudslay & Cotterell, 2021; Elazar et al., 2021), our work seeks to address, at least partially, the shortcomings most associated with probing individual neurons (Antverg & Belinkov, 2021). Methodologically, sparse probing is situated within the broader category of probing methods (Hennigen et al., 2020; Voita & Titov, 2020; Pimentel et al., 2020; Cao et al., 2021), which itself is just one paradigm among other localization techniques (Bau et al., 2018; Dhamdhere et al., 2018; Amjad et al., 2021; De Cao et al., 2021; Durrani et al., 2022; Goldowsky-Dill et al., 2023).

**Mechanistic Interpretability**   Philosophically, our work is motivated by the growing field of mechanistic interpretability (MI) (Olah et al., 2020b; Elhage et al., 2021; Casper et al.). MI concerns itself with rigorously understanding the learned algorithms (circuits) utilized by neural networks (Olsson et al., 2022; Wang et al., 2022a; Nanda et al., 2023; Chughtai et al., 2023) in the hope of maintaining oversight and diagnosing failures of increasingly capable models (Ngo et al., 2023). Although research on reverse-engineering specific neurons in LLMs is limited, (Geva et al., 2020) proposed—and later refined (Geva et al., 2022)—the hypothesis that feed-forward layers function as key-value memories, responding to specific input features (keys) and updating the output vocabulary distribution accordingly (Dar et al., 2022). These results are further inspired by earlier work in vision models (Bau et al., 2017; Kim et al., 2018). However, given the differences in scale, training objective, modality, and architecture between vision models and LLMs, it is unclear which of these results we should expect to transfer.

**Superposition** Perhaps the most significant obstacle to interpreting neurons in LLMs, and consequently, the success of MI as a whole, is the phenomenon of superposition. As first observed by (Arora et al., 2018), superposition involves compressing multiple features into a smaller number of dimensions (Elhage et al., 2022b). Recent research has investigated when superposition occurs (Elhage et al., 2022b; Scherlis et al., 2022), how to design models to have less of it (Jermyn et al., 2022; Elhage et al., 2022a), and how to extract features in spite of it (Sharkey et al., 2022). Many of the underlying mathematical intuitions rely on prior work on compressed sensing (Donoho, 2006). Moreover, similar questions have been studied elsewhere in machine learning within the disentanglement literature (Bengio et al., 2013; Chen et al., 2016; Kim & Mnih, 2018).

**Connections to Neuroscience** In a promising demonstration of consilience, our previous discussion of superposition has striking analogues with coding theory from biological neuroscience—the study of how neurons in the brain map to sensory stimulus (Rieke et al., 1999; Olshausen & Field, 1997; Barlow, 2001). On one extreme, local coding theory posits the existence of "monosemantic" biological neurons which respond to a very specific stimulus (e.g., pictures of Jennifer Aniston (Quiroga et al., 2005)). Superposition is then analogous to sparse coding where a subset of neurons encode some feature about the input (Olshausen & Field, 1997; Vinje & Gallant, 2000). Finally population coding theorizes the responses of a whole brain region are relevant, analogous to the computations of a full layer being required to compute or represent a feature (Pouget et al., 2000; Panzeri et al., 2015).

## 3 Sparse Probing

### 3.1 Preliminaries

**Transformers** We restrict our scope to transformer-based generative pre-trained (GPT) language models (Radford et al., 2018) that currently power the most capable AI systems (Bubeck et al., 2023). Given an input sequence of tokens $x = [x_1, \ldots, x_t] \in \mathcal{X}$ from the vocabulary, a model $\mathcal{M} : \mathcal{X} \to \mathcal{Y}$ outputs a probability distribution over the token vocabulary $\mathcal{V}$ to predict the next token in the sequence. $\mathcal{M}$ is parameterized by interleaving $L$ multi-head attention (MHA) layers and multi-layer perceptron (MLP) layers. The central object of the transformer is the residual stream, the sum of the output of all previous layers. Each MHA and MLP layer reads its input from the residual stream and writes its output by adding it to the stream. MHA layers are responsible for moving information between token positions, MLP layers apply pointwise nonlinearities to each token independently and therefore perform the majority of the feature extraction, and the residual stream acts as the communication channel between layers. In this study, we are primarily concerned with the MLP layers (accounting for $\approx \frac{2}{3}$ of total parameters) and investigate models with parallel attention (i.e., where the MHA layer and MLP layer are applied concurrently). For such models the value of the residual stream (also referred to as the hidden state) for token $t$ at layer $\ell$ is

$$h_t^{(\ell)} = h_t^{(\ell-1)} + \text{MHA}^{(\ell)}\left(\gamma\left(h_t^{(\ell-1)}\right)\right) + \text{MLP}^{(\ell)}\left(\gamma\left(h_t^{(\ell-1)}\right)\right) \tag{1}$$

where $\gamma$ is a layer normalization nonlinearity Ba et al. (2016). Our focus is the MLP layers given by

$$\text{MLP}^{(\ell)}(x) = W_{\text{proj}}^{(\ell)} \sigma\left(W_{fc}^{(\ell)} x + b_{fc}^{(\ell)}\right) + b_{\text{proj}}^{(\ell)} \tag{2}$$

where $W_{\text{proj}}, W_{fc}, b_{\text{proj}}, b_{\text{fc}}$ are learned weight matrices and biases, and $\sigma$ is a GeLU Hendrycks & Gimpel (2016). For a more detailed mathematical explanation of the transformer architecture we refer the reader to (Elhage et al., 2021).

**Probing** is a technique to localize where specific information resides in a model by training a simple classifier (a probe) to predict a labeled feature of the input using the internal activations of the model (Belinkov, 2022). More formally, we require a tokenized text dataset $X \in \mathcal{V}^{n \times T}$ (where $n$ is the number of sequences and $T$ is the context length) and an associated labeled dataset $D_{\text{probe}} = \{x_{jt}, z_{jt}\}$ which provides labels for a subset of the tokens (e.g., the tense of every verb). In our setting, we focus on the MLP neuron activations $a^{(\ell)} = \sigma(W_{fc}^{(l)}\gamma(h_t^{(l-1)}))$—the set of representations immediately after the elementwise nonlinearity—and

train a linear binary classifier $g_\ell(a_{jt}^{(\ell)}) = \hat{z}_{jt}$ to minimize a classification loss $\mathcal{L}(z_{jt}, \hat{z}_{jt})$ for each layer of the network.

**Key Concepts**  We make frequent reference to the concept of a *feature*. The field has not yet settled on a consensus definition (Elhage et al., 2022b), but for our purposes, we mean an interpretable property of the input that would be recognizable to most humans. A *monosemantic* neuron is a neuron which activates for exactly one feature, though this can also be subtle depending on what one is willing to count as one feature versus a composition of related features. In contrast, a *polysemantic* neuron is a neuron which activates for multiple unrelated features. Polysemanticity is a consequence of *superposition*, the phenomenon of representing $n$ features with $d < n$ dimensions. We focus on applying sparse probing to the activations of the MLP layers because these activations form a *privileged basis* (Elhage et al., 2021). That is, because applying an elementwise nonlinearity breaks a rotational invariance of the representations, it is more likely that the basis dimensions (each neuron) are independently meaningful. Without a privileged basis, as in the residual stream, there is no reason for features to be basis aligned, and therefore no reason to expect sparsity (modulo optimization quirks (Elhage et al., 2023)). Finally, superposition manifests differently in the residual stream than in the neuron activations because of this privileged basis; we discuss this further in A.9.

## 3.2  Sparse Feature Selection Methods

Rather than just localize a feature to a particular layer, we attempt to identify a single neuron or a sparse subset of neurons which fire if (and ideally only if) the feature is present in the input. This more granular localization can be accomplished by training a $k$-sparse probe, a linear classifier with at most $k$ non-zero coefficients. The problem becomes, which of the $\binom{d}{k}$ possible neuron subsets are most predictive? While this is an $\mathcal{NP}$-hard combinatorial optimization problem, there exist a number of fast heuristics and tractable optimal algorithms (Tillmann et al., 2021).

In probing, this is typically formalized as a ranking problem (Antverg & Belinkov, 2021), where neurons are individually scored by some importance measure and then rank ordered to include the top $k$. Given a binary classification dataset, natural scoring rules that have been explored are the absolute difference between class means for each neuron (Antverg & Belinkov, 2021), the mutual information between each neuron and the labels (Ross, 2014; Voita & Titov, 2020; Pimentel et al., 2020), or the absolute magnitude of the coefficients of a dense probe trained with $l_1$ regularization (Dalvi et al., 2019).

We propose two additional techniques: adaptive thresholding and optimal sparse probing (OSP). OSP leverages recent advantages in sparse prediction solution techniques to train a $k$-sparse classifier to provable optimality using a tractable cutting plane algorithm (Bertsimas et al., 2021), though this is only feasible for smaller values of $k$. For larger ranges of $k$, we use adaptive thresholding to train a series of classifiers that iteratively decrease the value of $k$, where at each step $t$ we retrain a probe to only use the top $k_t$ neurons with highest coefficient magnitude from the previous $k_{t-1}$-sparse probe. For the sake of computational efficiency, in both methods we perform an initial filtering step to take the top neurons with maximum mean difference.

For all methods, we retrain a logistic regression probe for the $k$ neurons selected. A comparison of these different methods is given in B.5. Additionally, while we focus on classification, all of these methods have a straightforward generalization to continuous targets.

## 3.3  Probing in Practice

**Constructing Probe Datasets**  An informative probing experiment begins with designing a suitable probing dataset. To illustrate a common issue, take the case of probing for the `is_politician` feature. If the dataset just includes the names of people labeled as politicians or not, the trained probe will not be able to distinguish the model's `is_politician` feature from the `is_political` feature which fires for all political content (e.g., "The Democratic Party"). On the other extreme, if the dataset contains just the names of politicians and random tokens, the probe won't distinguish `is_politician` from `is_person` as very few random tokens concern non-politician people. In other words, there is a general tension in how to shape the negative examples in the dataset to create the most *conceptual separation* between the true feature of interest and all possible correlates.

Another issue arises when the feature of interest is a property of multiple tokens or a full sequence of tokens (e.g., `is_python_comment` or `is_french`). If the relevant neuron fires on every token in the feature sequence, one could just sample tokens randomly, but there might be more specific conditions on when a neuron fires that cannot be known *a priori*[1]. A solution to this is to perform an elementwise aggregation (e.g., mean or max) of the activations over the token span of each occurrence of the feature. For example, for a dataset of sequences in many different languages, the input to our probe could be the average activation vector for each sequence, with the target being the language of the sequence. Unfortunately, this gives a somewhat weaker result, as this process is not able to distinguish the `is_french_noun` feature from the `is_french_verb` feature and thus requires further analysis to interpret correctly.

There are additional subtleties having to do with feature granularity, rare features, and overlapping features (Ravichander et al., 2020) which we explore in Section 5.4. For all of these issues, the most appropriate recourse will depend on the researcher's intent and the nature of the feature being probed for.

**Evaluation and Interpretation**    To evaluate the performance of a probe, we compute the number of true positives (TP), false positives (FP), false negatives (FN), and true negatives (TN) of our binary classifier on an out-of-sample test set. We then calculate the precision (PR), recall (RE), and F1 score (F1):

$$PR = \frac{TP}{TP + FP} \qquad RE = \frac{TP}{TP + FN} \qquad F1 = \frac{2PR \times RE}{PR + RE}$$

We use the F1 score as our primary evaluation metrics to determine which neurons are most likely associated with the target feature given the asymmetric importance of the positive class[2]. Precision and recall give insight into how the selected neurons' implicit feature granularity compares to the feature being probed for. Low precision and high recall indicates either selected neurons are highly polysemantic or the model represents a more general feature than is being probed for. High precision and low recall of the probing classifier may indicate that the identified submodule represents a more specific feature than the feature being probed for (e.g. `is_french_noun` instead of `is_french`). We can adjust our preference for classifiers with high precision or high recall by modifying the threshold of the classifier or by tuning the class weights in the probe loss function; a higher class weight for the positive class (assuming it has fewer samples than the negative class) will prioritize recall over precision.

## 4    Empirical Overview

**Models**    We study EleutherAI's Pythia suite of autoregressive transformer language models (Biderman et al., 2023) [3]. These models are fairly standard GPT variants that use parallel attention and rotary positional encodings and were trained on The Pile (Gao et al., 2020). In particular, we run experiments on the 7 models ranging from 70M to 6.9B parameters; full model hyperparameters are given in Table 1.

**Data**    We study ten different feature collections: the natural language of Europarl documents, the programming language of Github source files, the data source of documents from The Pile, the part-of-speech and grammatical dependency of individual tokens, morphological features of tokens (e.g., verb tense), plain text features of tokens (e.g., whitespace or capitalization), the presence of specific compound words, LaTeX features in ArXiv documents, and a number of factual features associated with people (e.g., gender, occupation). Full descriptions of datasets, their construction, and summary statistics are available in B.2. Categorical features are converted into separate one-versus-all binary features for over 100 total binary classification tasks.

**Experiments**    For each combination of model, feature, layer, and sparse feature selection methods, we train probes for a range of values for $k$ and report classification performance on a held-out test set. We compare the feature selection methods in B.5 and illustrate the relationship between model size and sparsity

---

[1]As an example, in the case of factual recall, the results from (Meng et al., 2022) suggest that MLP neurons are especially important for the last token of a multi-token subject.

[2]We also measure Matthew's Correlation Coefficient (MCC), a balanced accuracy metric, and the results are largely the same.

[3]https://github.com/EleutherAI/pythia; unfortunately, our experiments were performed with the V0 suite of models which have recently been updated.

in Figure 7. For neurons identified as being especially relevant, we save the activations over larger text data sets and perform further analyses described throughout Section 5. All code and data are available at `https://github.com/wesg52/sparse-probing-paper`.

## 5 Case Studies

We conduct a series of more detailed case studies to carefully study the behavior of individual neurons, while also illustrating the challenges that pose barriers to further progress. Although we zoom in on very narrow examples, we found many neurons of the same category in our probing experiments and believe these examples are representative of broader neuron families (Olah et al., 2020b) that exist in all LLMs.

### 5.1 Superposition in the Wild: Compound Word Neurons

The token vocabulary is a fairly unnatural symbolic language to perform all of the linguistic and conceptual processing required of a language model. For instance, compound words like "social security" are treated as two separate tokens, despite meaning something very different when the tokens appear together versus apart. Moreover, quirks of spacing and capitalization frequently cause words to be broken up (e.g., despite " Harvard" being a single token, when not preceded by a space "Harvard" gets tokenized into "Har" and "vard"). Based on many such examples, (Elhage et al., 2022a) hypothesize that a primary function of the early layer neurons is to "de-tokenize" the raw tokens into a sequence of more useful abstractions. However, this *pseudo-vocabulary* might be extremely large (e.g., all common $n$-grams) making it a natural candidate for being represented in superposition. To investigate this, we probe for neurons which respond to 21 different compound words, where the first and second words mean something quite different when appearing separately versus together (e.g., "prime factors").

After probing for neurons which activate for specific compound words `XY` while not firing on any bigrams of the form `XZ` or `WY` $\forall$ `W,Z` $\in \mathcal{V}$, `Z`$\neq$`Y`, `W`$\neq$`X` for each compound word we found *many* individual neurons which were almost perfectly discriminating. However, after inspecting the activations across a much wider text corpus, we observe these neurons activate for a huge variety of unrelated $n$-grams (see Figure 2(a)), a classic example of the well known phenomena of polysemanticity (Olah et al., 2020b; Mu & Andreas, 2020; Elhage et al., 2022a). Superposition implies polysemanticity by the pigeonhole principle—if a layer of $n$ neurons reacts to a set of $m \gg n$ features, then neurons must (on average) respond to more than one feature. This example also underscores the dangers of "interpretability illusions" caused by interpreting neurons using just the maximum activating dataset examples (Bolukbasi et al., 2021). A researcher who just looked at the top 20 activating examples would be blind to all of the additional complexity of neuron `70M.L1.N111`, with Figure 2(a) still only scratching the surface[4]. While inconvenient for interpretability researchers, polysemanticity is also problematic for the model, as it causes interference between different features (Elhage et al., 2022b). That is, if `70M.L1.N111` fires, the model gets mixed signals that both the "prime **factors**" feature and the "International C**oven**" feature are present.

However, detecting $n$-grams in particular (and hence constructing the hypothesized pseudo-vocabulary), turns out to be a task particularly well suited for superposition (see A.6). In short, the model can take advantage of the fact that (for fixed $n$) exactly one possible $n$-gram out of $|\mathcal{V}|^n$ can occur at any time—that is, $n$-grams are mutually-exclusive binary features with respect to the current token of the input, despite coming from a massive set of possible features. While it would be impossible to dedicate a unique neuron to all possible $n$-grams, by leveraging the activations of multiple polysemantic neurons, each of which react to a bigram `XY` but with no other overlapping stimuli, the magnitude of the "true" feature gets boosted above all of the possible interfering features. As an example, in Figure 2(b), we show three neurons from the same layer which activate for the "social security" bigram while (mostly) not activating for bigrams with just one of the words (see the blue histograms being significantly to the right of the red and green histograms). However, because these neurons are polysemantic, there are many other inputs which cause them to activate, potentially even more so than the "social security" feature (see the orange histogram having a longer tail than

---

[4]This figure was generated with 300 million tokens from the Pile Test set but when run on 10 billion tokens of the Pile train set 16 of the top 20 dataset examples show the neuron activating on the "he" of German words starting with "Schönhe". These maximum activating dataset examples can be viewed on Neuroscope(Nanda, 2022)

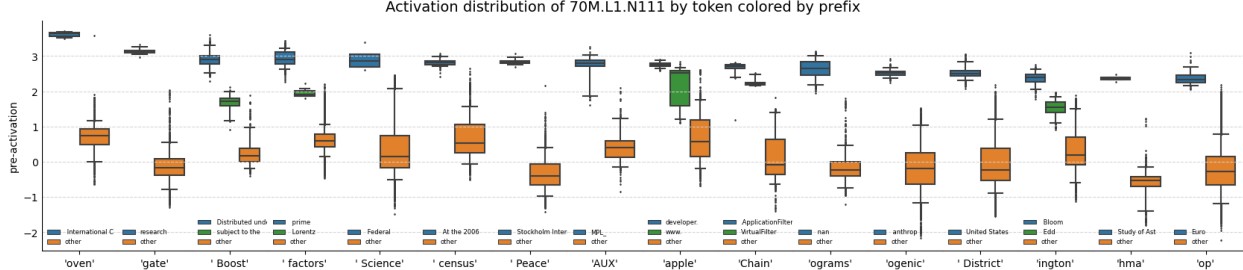

(a) Activations of a single polysemantic neuron on different tokens when preceded by specific stimuli.

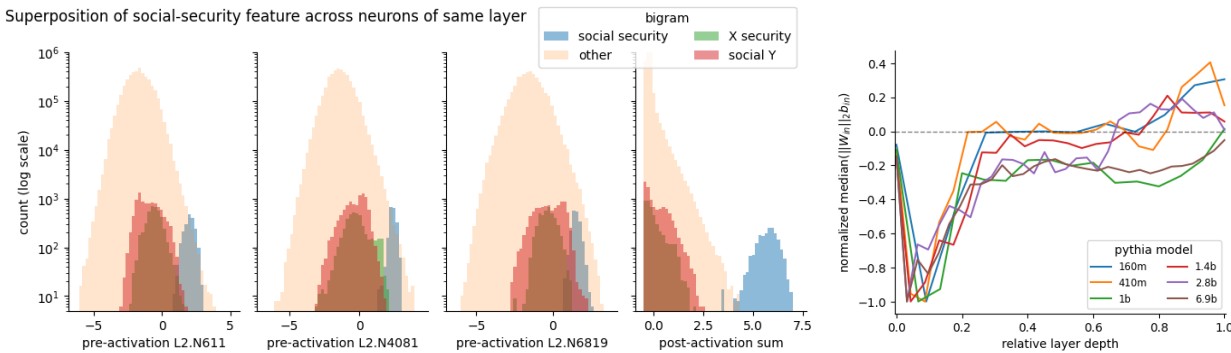

(b) Example of superposition in representing the compound word "social security." (c) Evidence of superposition in early layers

Figure 2: Summary of our key results of superposition. Polysemantic neurons firing on many diverse stimuli (a) are necessary to implement superposition (b) to adequately represent many more possible $n$-grams than dimensions. The natural mechanism for implementing superposition in toy models leverages large weight norms and negative biases(Elhage et al., 2022b) (see Section C for additional motivation and discussion), which we observe to be most associated with early layers (c).

the blue histograms). Despite this polysemanticity, by summing the activations across the three neurons, we achieve nearly perfect separation between the total activation magnitude of the "social security" feature and *all* other observed token combinations! We include examples for all 21 compounds words from layers 1-3 of Pythia 1B in Figures 9, 10, and 11, and further results on basis alignment in A.10 and Figures 13 and 14. We believe this to be first example of neuron superposition exhibited "in the wild" in real LLMs.

We believe that this computational motif—merging individual tokens into more semantically meaningful $n$-grams in the pseudo-vocabulary via a linear combination of massively polysemantic neurons—is one of the primary function of early layers. As a supporting line of evidence, we observe that a natural idealized model of $n$-gram recovery in superposition predicts a mechanistic fingerprint of superposition: the presence of large input weight norms and large negative input biases. We provide an idealised construction in Section C for how an arbitrary number of features can be compressed into two linear dimensions, similar to the construction in (Elhage et al., 2022b). When we measure the product of the input weight norm and bias for each neuron in every layer and every model, we observe a striking difference in the early[5] layers, exactly in line with our conceptual argument (see Figure 2(c) for a summary and Figure 8 for the full distributions, and Section C for additional motivation of the metric and discussion).

We conclude by noting that, in another encouraging display of consilience, the phenomenon of different regions of the model employing different coding strategies for different functions bears resemblance to the diverse coding strategies employed by biological neural networks in different brain regions with different functional roles (Olshausen & Field, 1996; Hromádka et al., 2008; Johnson & Leon, 2000; Leutgeb et al., 2005). We expect further research connecting local mechanisms to macroscopic structure to be particularly fruitful.

---

[5]Pythia models use parallel attention, so layer 0 is only a function of the current token and not subject to the $n$-gram analysis.

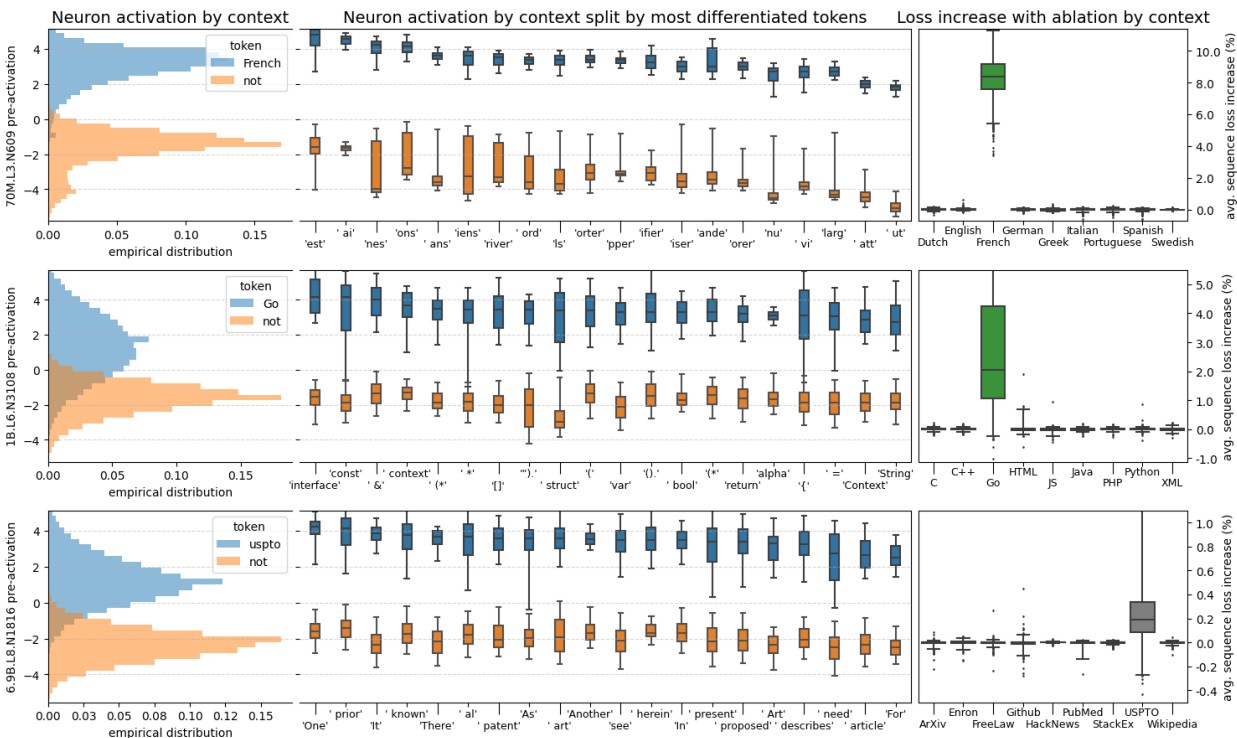

Figure 3: Analysis of individual context neurons that activate on tokens in French (top), in Go code (middle), and US Patent office documents (bottom). We show the distribution of activation when in and not in the target context (left) in addition to the tokens with average activation most distinguished by the neuron (middle). Finally we report the increase in sequence loss across contexts when the neuron is ablated (right).

## 5.2  Context Neurons: a Monosemantic Neuron Family

Given the potential benefits of superposition, it is reasonable to ask if *any* features are represented monose-mantically, and if so, why? We hypothesized that a likely candidate would be *context features*—high-level descriptions of all (or most) tokens within a sequence (e.g. `is_french` or `is_python_code`). Such features seem quite important, to the point that avoiding interference is worth a full neuron, while also being a higher-level property that may or may not be mutually exclusive or binary, making it harder to represent in superposition.

In particular, we probe on the language of natural language sequences from the Europarl dataset (Koehn, 2005), the programming language of different Github source files, and the data subset of The Pile from which a sequence originated. We probe for these features using mean aggregation—training a classifier on the averaged activations over each sequence to predict the sequence label (e.g., `is_french` or `is_python`). Figure 3 depicts a sample of our results, illustrating the existence of highly specialized context neurons that activate approximately only when a token is in a specific context[6] (with 21 additional examples in Figures 15, 16, and 17). To better understand the function of these neurons, for each token, we study the distribution of activations broken down by context (e.g., the activation of "return" in every programming language). Sorting by the largest differences (middle panel), we observe that one explanation for these neurons' roles is token disambiguation. For example, many programming languages have a `return` keyword, but neuron `1B.L6.N3108` activates on `return` if and only if it is in the context of Go code.

To further support our interpretation and gain more insight into function, we conduct ablation experiments comparing the language modeling loss of the base model to the loss of the model with the identified neuron fixed

---

[6]By recording the activation of every token, we eliminate the concern of finding an `is_french_noun` neuron instead of an `is_french` neuron.

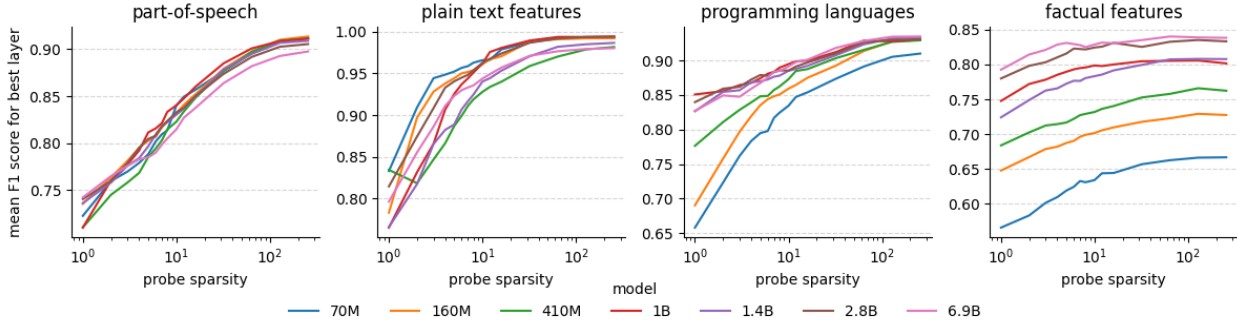

(a) Probe sparsity versus classification performance for different feature types and model sizes.

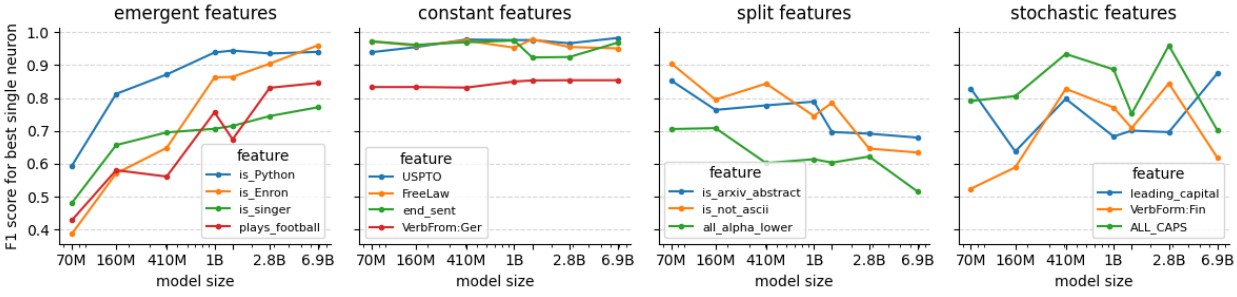

(b) Classification performance of most feature-aligned neuron by model size for different feature classes.

Figure 4: Summary of results on model scale. Representational sparsity increases on average with scale (top) but individual features obey different scaling dynamics (bottom).

to 0 for all tokens in every sequence (right panel). As anticipated, ablating the context neurons significantly degrades language modeling performance within the relevant context, while leaving other contexts nearly unaffected. The impact, however, heavily depends on the model size—in the 70M parameter model ($\approx 12$k neurons), ablating a single neuron causes an average loss increase of 8% per French sequence, while in the 6.9B model ($\approx$524k neurons), ablating one neuron results in only a 0.2% increase in loss.

While these neurons appear to be genuinely monosemantic, we emphasize that it is extremely difficult to prove this. Doing so requires answering thorny ontological questions (what *is* French?) and then efficiently searching through a dataset of billions of tokens to verify the neuron implements the answer. Moreover, the "true" feature the neuron responds to might be quite subtle. For instance, many of the code neurons do not fire when in a code comment, whereas the French neuron will fire in a non-French context on a French name or other token strongly associated to France[7]. In some sense, these exceptions prove the rule of the primary role of the neuron, but highlight the difficulty of mapping human features to the ontology of the network. Perhaps most challenging though is ruling out the existence of very rare features the neuron also responds to. Sufficiently rare features will likely be undetectable from random samples or summary statistics—even when broken down by token—and therefore require manually[8] examining max activating dataset examples in subdistributions where we don't expect to find much French text.

### 5.3 Effects of Scale: Quantization and Splitting

Given the importance of scale in LLMs, we turn our attention to studying the relationship between model size and the sparsity of representations, and the dynamics that drive this relationship. For all of our feature datasets described in B.2, we train a series of probes sweeping the value of $k$ from 256 to 1 using adaptive thresholding. For sake of summarization, we report the maximum out-of-sample F1 value over the layers for

---

[7]To further explore this, we encourage the reader to explore max activating dataset examples in Neuroscope(Nanda, 2022)
[8]This is an area where we expect AI assisted interpretability to be particularly useful.

each model, while averaging over the features within the collection (see Figure 4(a) for a sample and Figure 7 for full results with random baselines).

While some features appear to be more sparsely represented with scale (e.g., programming languages and factual features), we were surprised by how consistent others were (e.g., part-of-speech and compound words). In fact, for some of the simplest plain-text features, the smallest models seem to actually implement greater sparsity. When analyzing individual features on a per neuron basis, several general patterns become more apparent (Figure 4(b)). We believe there are two main dynamics driving these results: the quantization model of scaling (Michaud et al., 2023) and neuron splitting (Elhage et al., 2022a).

In particular, the quantization hypothesis posits that there exists a natural ordering of features learned by a model with increasing scale based on how loss reducing they are, with larger models able to learn a longer tail of increasingly rare features (Michaud et al., 2023). In our context, this suggests that features like part-of-speech or compound words are within the feature set learned by even small models, and are represented with very similar sparsity patterns. In contrast, factual features and some (but not all) contextual features only get represented by single neurons at sufficient scales. As a countervailing force, with increasing scale the model can dedicate multiple neurons to represent more granular features that previously comprised one coarser feature. Consider the `ALL_CAPS` feature; while it would likely be advantageous to have dedicated circuitry for representing this feature in all models, a larger model might have dedicated neurons for all of the particular reasons a token might be in all capital letters (e.g., an abbreviation, a constant in python, shouting on the internet, etc.), eliminating the need to have just one more coarse grained representation. The result, as viewed from a probing experiment, is less sparsity, not more.

### 5.4    Refining and Verifying Interpretations

The result of a sparse probing experiment is a set of $k$ neurons with a collection of classification performance metrics, usually for a range of values of $k$. We illustrate what conclusions might follow, simple techniques to refine and corroborate such conclusions, and confounding factors to be wary of.

One confounding factor is an ambiguity between superposition and *composition* (Mu & Andreas, 2020; Olah, 2023). If a (layer, feature) pair has a 1-sparse probe with poor accuracy and a $k$-sparse probe with high accuracy, it is temping to take this as evidence of superposition, but it is also consistent with a feature being coarse or compositional—being either the union or intersection of multiple independent features. As an example, consider an `is_athlete` feature. Such a feature could either be represented by a single neuron, a superposition of polysemantic neurons, or as the union of neurons for specific types of athletes, as those shown in Figure 5(a). While we can distinguish a feature union from superposition by analyzing the activation co-occurrence (only one athlete type neuron activates for each athlete), this reasoning does not work for a true compositional feature. For example, we also found individual neurons which coded for a person's gender and whether they are alive or not (Figure 20). It is likely more natural for a model to represent the `is_living_female_soccer_player` feature as simply a composition of three different neuron aligned-features `is_living`, `is_female`, and `plays_soccer`, which would nevertheless have low 1-sparse accuracy but high 3-sparse accuracy. Of course, this also assumes that these neurons are all in the same layer. However, models can, and almost certainly do, leverage superposition and composition across multiple layers, a subtlety which we blissfully ignore and leave for future work.

For the remaining discussion, we consider individual neurons identified by 1-sparse probes—how do we interpret such neurons? The most basic analysis is to simply inspect the maximum activating dataset examples (Nanda, 2022), though this is subject to interpretability illusions (Bolukbasi et al., 2021). A slightly more robust analysis involves computing the average activation for each token in the vocabulary. Doing so for our athlete neurons in Figure 5(a) reveals these neurons are perhaps better thought of as generalized sport neurons which activate for all tokens generally having to do with a particular sport, including the names of athletes (with the notable exception of the hockey neuron which appears to be a Canadian neuron).

In addition to analyzing a neuron via the input, one can also gain insight by analyzing the output; in particular, one can compute a neuron's effect on the output logits by simply considering the product of the unembedding matrix and the neuron output weight (Dar et al., 2022). Doing so for neurons which always activate on punctuation ending a sentence, we find that the output probability of most uppercase tokens gets

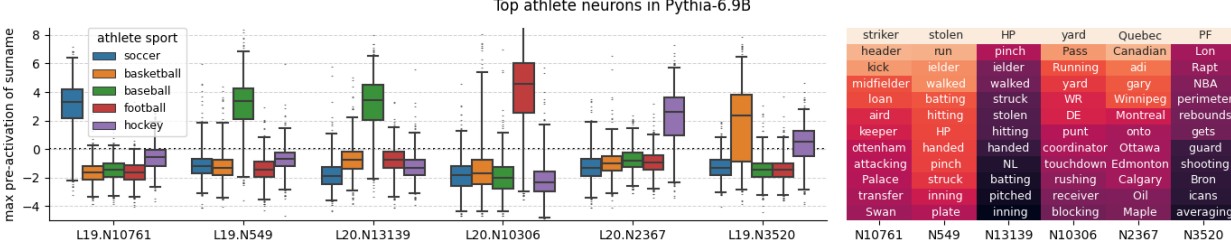

(a) Neurons which activate for the names of specific types of athletes (left) turn out be more general sport neurons when analyzing the top per-token average activations (right).

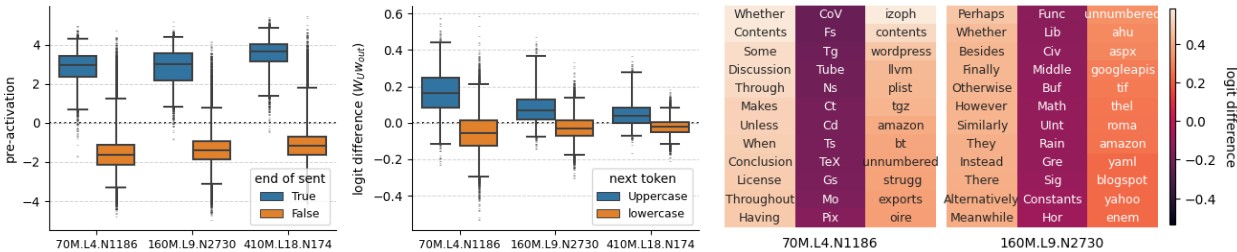

(b) End of sentence neurons: analyzing the effect on the output vocabulary can corroborate and refine neuron interpretations. For token heatmaps (right), the columns correspond to capital tokens with max increased logits, capital tokens with max decrease, and lowercase tokens with max increase, respectively.

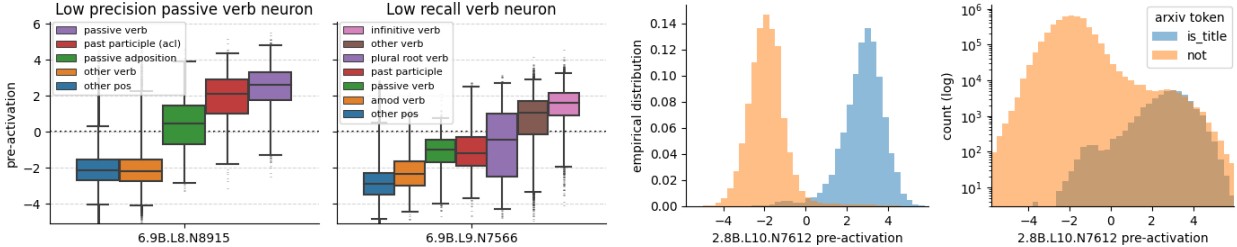

(c) Precision and recall metrics can guide further analysis.     (d) Distributional statistics can hide rare features.

Figure 5: Examples of analyses used to refine and support neuron interpretations.

increased while the probability of most lowercase tokens gets decreased (Figure 5(b)). Inspecting specific examples is often constructive; for instance, the highest increasing lowercase tokens are words like "amazon" or "wordpress," indicating such neurons are also useful in constructing URLs. For uppercase tokens, those with the highest increase in probability correspond to words like "Finally" or "However" while the most decreased tokens are those representing atomic elements, proper nouns, or CamelCase strings.

Most of the neurons discussed thus far have had excellent overall classification performance; what happens if there does not exist such a crisp fit? As discussed in Section 3.3, classification performance can be further decomposed into precision (sensitive to false positives) and recall (sensitive false negatives). In our context, high recall and low precision with respect to a specific feature potentially suggests that a neuron represents a less granular feature than the feature manifest in the probe dataset [9]; low recall and high precision suggest a neuron represents a more specific feature. As an example, Figure 5(c) depicts the activations for a high-recall-low-precision neuron identified when probing for `is_passive_verb` and a low-recall-high-precision neuron identified when probing for `is_verb` in Pythia 6.9B. When we analyzed these neurons in more detail, we find that in addition to activating on all occurrences of passive verbs, `L8.N8915` also activates on adjacent adposition tokens, and past participles when in an adnominal clause. Almost symmetrically, `L9.N7566` activates on most verbs, especially infinitive verbs, but not passive verbs, adjectival modifiers, plural root

---

[9]It is also consistent with superposition. Indeed, all of the compound word neurons had higher recall than precision.

verbs, or past participles in certain dependency roles. These examples illustrate the more general point that it should not be assumed that a model will learn to represent features in an ontology convenient or familiar to humans.

Of course, precision and recall are only defined with respect to the labels of a probing dataset, which may themselves contain imbalances or spurious correlations that confound the results. Perhaps most common, especially when constructing a dataset from scratch, is the problem of asymmetric sampling and rare features. Consider a probing dataset for an `is_arxiv_paper_title` feature (Figure 5(d)). Without a more specific hypothesis on the relevant negative examples, one is likely to simply sample random non-title tokens (or at least weight all non-title examples the same). However, when the space of negatives is large (e.g., all non-title tokens), at a distribution level, all rare features get drowned out. Hence, when looking at summary or distributional statistics, the results can look impressive (Figure 5(d); the neuron had F1 > 0.95 for `is_title`), but when looking at raw counts (Figure 5(d) right) one observes that the `is_title` feature actually explains less than half of the activations. Though, upon inspection, we find that the rest of the activations are for *section* titles!

# 6 Discussion

## 6.1 Strengths and Weaknesses of Sparse Probing

The primary use case of sparse probing is to quickly and precisely localize neurons relevant to a specific feature or concept, while naturally accounting for superposition and composition. The speed and precision of recovery is in contrast to gradient-based (De Cao et al., 2021) and causal-intervention based methods (Meng et al., 2022), which are too slow or coarse-grained to be used for features requiring precise localization within large models. The feature specificity contrasts with sparse autoencoding based methods which aim to recover *all* features stored in the model, but in an unsupervised manner and without attributing semantic meaning to each feature (Sharkey et al., 2022). Having probes with optimality guarantees further addresses the pitfall raised by (Antverg & Belinkov, 2021) regarding the conflation of classification quality and ranking quality when analyzing individual neurons with probes. Moreover, sparse probes are designed to be of minimum capacity, mitigating the concern that the probe is powerful enough to construct its own representation to learn the task (Hewitt & Liang, 2019). While probing requires a supervised dataset, once constructed, it can be reused for interpreting any model (modulo features regarding tokenization), enabling investigation into the universality of learned circuits (Olah et al., 2020b; Chughtai et al., 2023) and the natural abstractions hypothesis (Chan et al., 2022). By design, sparse probing is particularly well suited for studying superposition, and can be used to automatically test the effect of architectural changes on the frequency of polysemanticty and superposition, as opposed to relying on human evaluations as in (Elhage et al., 2022a).

However, sparse probing also inherits many of the weaknesses of the general probing paradigm (Hewitt & Liang, 2019; Donnelly & Roegiest, 2019; Ravichander et al., 2020; Maudslay & Cotterell, 2021; Elazar et al., 2021). In short, the results from a probing experiment do not allow for drawing especially strong conclusions without more detailed secondary analysis on the identified neurons. Probing offers limited insight into causation, and is highly sensitive to implementation details, anomalies, misspecifications, and spurious correlations within the probing dataset. For interpretability specifically, sparse probes cannot detect features built up over multiple layers or easily distinguish between features in superposition versus features represented as the union of multiple independent and more granular features (Mu & Andreas, 2020). In seeking the sparsest classifier, sparse probing may also fail to select important neurons that are redundant within the probing dataset, requiring the use of iterative pruning to enumerate all important neurons. Multi-token features require special processing, often in the form of aggregations that can further weaken the specificity of the result. Finally, depending on how much representations change with scale (Michaud et al., 2023), larger models may utilize more specific features (Elhage et al., 2022a), hampering the transferability of datasets between different model scales.

### 6.2  Strengths and Weaknesses of Empirical Findings

In light of the limitations of probing, we attempted to corroborate every case study with independent evidence, including theoretical predictions, ablations, and vocabulary analyses. We find our evidence compelling, although not always conclusive. In particular, we believe we have provided the clearest evidence to date of superposition, monosemantic neurons, and polysemantic neurons in full scale language models. Furthermore, by demonstrating this behavior in seven different models spanning two orders of magnitude in size while exploring over a hundred features, we believe our basic insights are likely to be general and to transfer to current frontier models like GPT-4.

However, much of our analysis is *ad hoc*, tailored to the specific feature being investigated, and requires substantial researcher effort to draw conclusions. While we explored models of varying size, they were all from the same model family and trained with the same data. We think it is unlikely our results are specific to the implementation details of the Pythia model suite, but we do not rule this out. Additionally, the largest model we studied is 6.9 billion parameters which is still more than order-of-magnitude off the frontier. Given the emergent abilities of LLMs with scale (Wei et al., 2022), it is possible our analysis misses a key dynamic underlying the success of the largest models. Moreover, our results are restricted to binary features and categorical features converted to binary features, and we are not confident that our insights will cleanly transfer to continuous features or cleverly encoded categorical features.

### 6.3  Implications

For interpretability researchers, our results support the conclusion that superposition is important for the success of models. Hence, attempts to remove it (Elhage et al., 2022a) are likely either hiding it or are unlikely to be competitive, but that the highest leverage interventions are perhaps best aimed at the early layers. For AI ethicists and legal scholars, our study of factual neurons point to an important avenue of further inquiry—understanding how neurons encoding protected attributes compute this feature and affect downstream predictions of the model. For AI alignment scholars, our work highlights the potential of identifying safety-critical features and perhaps even manually intervening in computations to enhance our ability to steer models. While none of the features we probed for were especially critical, the Pile subset context neurons could be considered a minimal precursor for situational awareness (Ngo et al., 2023; Carlsmith, 2022), as these could be used to detect whether an input is from the training or test distribution.

### 6.4  Future Directions

We only scratched the surface of possible applications and experiments involving sparse probing. Scientifically, further understanding superposition—and how to cope with it—seems central to making progress on the ambitious version of mechanistic interpretability. As outlined in (Elhage et al., 2022b), this requires either designing architectures which do not use superposition or to take features out of superposition using a sparse coding like technique—both of which can be assisted with automatic sparse probing as a form of validation. While not explored here, it would also be possible to apply sparse probing to predicting properties of the *output* (e.g. `next_token_is_verb`), as opposed to properties of the input, to better understand the neurons most implicated in making specific types of predictions. With neurons identified in each experiment, it would be potentially insightful to also track how these neurons develop and change through time (Liu et al., 2021) as well as more carefully analyze how the set of neurons change with scale (Michaud et al., 2023), which is naturally enabled by the Pythia suite's (Biderman et al., 2023) checkpoints. In particular, sparse probing (or similar) is well suited to study neuron splitting in more detail by training probes to predict the value of a less granular neuron from the sum of a sparse set of more granular neurons.

Our primary motivation however, is in ensuring the development of safe AI systems. To further this goal, we envision the development of a large library of probing datasets—potentially with AI assistance—that capture features of particular relevance to bias, fairness, safety, and high-stakes decision making. In addition to automating evaluations of new models, having large and diverse supervised datasets will enable better evaluations of the next generation of unsupervised interpretability techniques (Sharkey et al., 2022; Burns et al., 2022) that will be needed to keep pace with AI progress.

## 7 Conclusion

Guided by sparse probing, we have found some of the cleanest examples of monosemanticty, polysemanticity, and superposition in language models "in the wild," and contributed practical guidance and conceptual clarification for how to interpret neurons in greater detail. More than any specific technical contribution, we hope to contribute to the general sense that ambitious interpretability *is* possible—that LLMs have a tremendous amount of rich structure that can and should be understood by humans. We believe this is most productively accomplished with an empirical approach more reminiscent of the natural sciences, such as biology or neuroscience, than the traditional experimental loop of ML. While such research can be hard to finish, it is easy to start, so we encourage the curious researcher to just start looking!

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

# A  Frequently Asked Questions

## A.1  Why expect ambitious interpretability to be possible or worthwhile?

We divide our answer into two parts: why it should be possible even in theory, and how we interpret the progress so far.

Although gradient descent has no reason or incentive to learn representations scrutable to humans, the same could be said of all the biological structures "learned" by natural selection. Yet biological structures are constrained by the laws of nature—an organism must make efficient use of limited energy and space, its genetic material is encoded in RNA and DNA, its functionality is encoded in proteins, and these proteins fit together into biological circuits. Although these biological circuits are complex, often idiosyncratic to organisms, and at times seem inscrutable, these constraints give us insight into their functioning and create common motifs and principles that can be reverse-engineered (Alon, 2009). In a similar way, neural networks are trying to achieve a complex tasks and have significant constraints, namely, the number of parameters, non-linearities and weight norm. And more broadly, the algorithms their architecture allows them to represent. By studying analogous structures of neural nets—both at a mathematical level (Elhage et al., 2021) and in the wild (Wang et al., 2022a)—it is possible to uncover some of these principles and motifs, and to start gaining a foothold in reverse-engineering these artificial circuits.

A potential source of pessimism for ambitious interpretability is the progress made thus far. Though we have gained some insights about model internals, most works in mechanistic interpretability have focused on small or toy models (Elhage et al., 2021; Nanda et al., 2023; Chughtai et al., 2023; Wang et al., 2022a; Cammarata et al., 2021), while frontier models become larger at a far faster pace (Brown et al., 2020; Chowdhery et al., 2022; OpenAI, 2023). While more scalable approaches to interpretability have often been shown to have results that are easy to misinterpret, or sometimes actively misleading (Adebayo et al., 2018; Bilodeau et al., 2022; Hase et al., 2023; Karimi et al., 2022). It is natural to look at this confusion and seeming incomprehensibility and to feel discouraged. Yet, natural structures were no doubt similarly incomprehensible to early pioneers in molecular biology, genetics, and neuroscience, exhibiting an emergent complexity that seemed irreducible. However, rather than claiming cells, genes, or brains were "uninterpretable," entire scientific disciplines emerged which have made great strides in understanding the core principles in sufficient detail to enable intervening, engineering, and controlling biological systems. We believe artificial neural networks can be fully interpreted—even reverse-engineered—but doing so requires a comparable amount of effort as interpreting biological ones. In many ways, this emerging *artificial neuroscience* is unusually amenable to the scientific method (Olah, 2021): it is possible to run arbitrary counterfactual experiments, iteration times are rapid, resource requirements are minimal, all with full observability and measurement precision equal to floating point machine precision. While ambitious, we think such an effort is paramount for addressing issues in the alignment and control of increasingly capable AI systems.

## A.2  What do you mean by a neuron?

The term **neuron** is sometimes used to refer to elements of any activation tensor, in the standard basis. Here, we instead reserve neuron to refer to the elements of the model activations immediately after an elementwise non-linearity: the post GELU activation in the hidden state of a transformer's MLP layer. We do *not* use neuron to refer to elements of a transformer's residual stream, a layer's output, or the key, query or value within an attention head. These are the output of a linear map, and so do not have a privileged basis (Elhage et al., 2021): we expect the model to function the same under an arbitrary rotation (modulo optimiser quirks (Elhage et al., 2023))

### A.3 What is the difference between polysemanticity, superposition, and distributed representations?

These are similar concepts with crucial differences, that are commonly confused. We propose the following schema:

**Superposition** is the phenomena when an activation represents more features than it has dimensions. Each feature is a direction in space, and as a consequence they *cannot* all be orthogonal.

**Polysemanticity** is when a neuron seems to represent multiple, unrelated concepts. This is in contrast to a **monosemantic** neuron that activates if and only if a certain feature is present.

**Distributed representations** are where a feature is represented as a linear combination of neurons, i.e., a direction that does *not* correspond to a single basis element. Also known as non-basis aligned features. Notably, this could be a fairly sparse distributed representation (e.g., using 2-5 neurons), fairly dense (e.g., using 10-20% of all neurons in the layer), to completely dense/not at all basis aligned.

Notably, polysemanticity and distributed representations are **local** notions; they can be demonstrated by studying individual neurons or features respectively. Superposition is a **global** phenomena, and requires identifying more features than neurons to conclusively show.

### A.4 What is the relationship between polysemanticity, superposition, and distributed representations?

A linear representation may be distributed because of rotation/skew (e.g., when lacking a privledged basis), composition, or superposition (Olah, 2023).

Superposition necessarily implies polysemantic neurons (and likely distributed representations), because there are more features than neurons! However, it is possible to have distributed representations and polysemanticity without superposition. There may be as many features as dimensions, but in some rotated basis not aligned with the neuron basis.

Indeed, without a privileged basis, this is what we would expect to observe even without superposition. The model has no incentive to align features with basis elements, and as we would expect, prior work (Bolukbasi et al., 2021) has found polysemanticity of residual stream basis elements. However, it is surprising that this should occur in the presence of an elementwise non-linearity. If a model is acting as a feature extractor, it is useful to represent features such that they can independently vary. Individual neurons have a separate GELU, but if multiple features share a neuron then they may significantly interfere with each other.

Moreover, we can have polysemantic neurons and superposition without a distributed representation if a neuron represents multiple features, but each of those features is *only* represented by that neuron. This is a form of superposition. The model will then need to have circuitry to disambiguate which feature activated the neuron, but we can imagine ways this could be efficiently implemented. A toy example: a model could have 100 neurons, each of which represents both some feature of Python code and some feature of romance novels. Python code and romance novels are mutually exclusive contextual features that could each already be computed and represented in the residual stream, which can then be used to disambiguate which feature each neuron activated for. Though this could be considered a certain kind of distributed representation where a feature is represented as a linear combination of its neuron and the romance/Python feature.

Our interpretation of our results is that this is further evidence that, at least in early layers, models engage in superposition involving both polysemanticity and distributed representations.

### A.5 What is the difference between faceted and polysemantic neurons?

A basic approach to detecting polysemanticity is to apply clustering to the activations of texts that strongly activate a neuron (e.g., the residual stream immediately before the MLP layer (Black et al., 2022)): if each cluster has some shared semantic meaning, then if there are multiple clusters the neuron is seemingly polysemantic! And indeed, if there is a single clear cluster, that is evidence of monosemanticity. However, there are two distinct phenomena here: related and unrelated clusters. A neuron is **faceted**(Nguyen et al., 2016) if it activates for multiple different things with shared meaning, e.g., a cutlery neuron activating for

knives, forks and spoons. While a **polysemantic** neuron activates for clusters without a shared meaning, e.g., dice and poetry. Unfortunately, this is currently a subjective definition: what does it *mean* to have shared meaning? We see formalizing these intuitive concepts as a promising area of future work.

### A.6   What are the two different kinds of interference, and how do they affect superposition?

The phenomena of superposition arises because models can decrease loss by representing more features, yet having non-orthogonal features introduces interference which increases loss. This creates a pareto-frontier trading off the two for each feature and each model, and this determines the representations a model learns. Crucially, interference between two features A and B decomposes into two conceptually different forms of interference (Nanda, 2023a): **alternating interference** where A is present, B is not present (or vice versa), and the model needs to tell that A is present and B is not, and **simultaneous interference** where A and B are present, but the model needs to tell that A and B are both present, but that, for example A is not present at twice strength. To understand the difference, let's consider the case where A and B are binary features corresponding to different directions of unit norm.

Alternating interference is fundamentally about distinguishing high activations (feature A is on) from low activations (feature B is on, feature A is off, but its direction is not the same as feature A's direction), which is fairly straightforwards with GELUs and softmaxes. But simultaneous interference is fundamentally about treating medium activations (feature A is on, feature B is off) as the same as high activations (feature A is on and feature B is on) but different from low activations (feature A is off and feature B is on, or both are off). This is much harder to do with GELUs or softmaxes.

In toy models, models seem to avoid superposition with high simultaneous interference (e.g., correlated features), but tolerate high alternating interference (e.g., anti-correlated features). Notably, (Elhage et al., 2022b) observed that independent features occurring with probability $p$ engage in more superposition when $p$ is lower - the probability of alternating interference is linear in $p$ while simultaneous is quadratic in $p$. The expected loss reduction from representing a feature is also linear in $p$, suggesting that whether or not a model engages in superposition is predominantly driven by the costs of simultaneous interference, not alternating. For sufficiently rare and uncorrelated features simultaneous interference is a non-issue.

Mutually exclusive features like $n$-grams are an extreme case where simultaneous interference can never occur, and so are well suited to superposition.

### A.7   How does the type of feature affect how it might be expressed in superposition?

Three notable categories of features are **continuous** (e.g., the height of an object), **binary** (e.g., whether 'social security' is present) and **categorical** (e.g., whether an object is blue, red or yellow). Categorical features can be represented as one-hot encoded binary features, so it is most instructive to compare binary features and continuous features. We argue that interference is significantly greater for continuous features than binary features—intuitively alternating interference will prevent us from distinguishing small values of the true feature and large values of an interfering feature.

This effect has been observed in the literature when studying toy models. (Elhage et al., 2022b) studied continuous variables and found superposition tending not to compress more than 5 features into two dimensions (and with significant interference), while (Henighan et al., 2023) studied binary variables (memorization) and found that models could compress significantly more features into two dimensions with minimal performance lost (though this is confounded, as memorization is also mutually exclusive).

### A.8   Why might we expect $n$-gram detection in particular to use superposition?

$n$-gram detection is unusually well suited to superposition for several reasons. $n$-grams are mutually exclusive ('social security' cannot co-occur with 'prime factors', for example), and are binary variables.

Moreover, $n$-grams are an example of a simple feature: to detect 'social security', a neuron need only compose with a previous token head to detect ' social' and with the current token embedding to detect ' security', which will create a sharp decision boundary: a significant gap between the smallest neuron activation when

the feature is present and the largest neuron activation when the feature is not present. This means that for downstream computation the ' social security' feature can be used with minimal noise from false positives. In contrast, complex or subtle features like whether a text is in French may require compiling many tiny pieces of evidence, creating an unclear decision boundary between true and false. In this way, despite the ground truth being a binary feature, models may think of complex and nuanced features as continuous variables associated with the probability of being present, making these features harder to store in superposition.

### A.9 What is the difference between residual stream and neuron superposition?

Residual stream superposition is an example of **representational superposition** (sometimes called bottleneck superposition) and is fundamentally about compression. That is, mapping a high dimensional space to a low dimension space such that the model can recover the compressed features at a later point in the network. This also implies that the model has already computed the feature, and the superposition is just for the purposes of storage. In addition to occurring in the residual stream, we also expect the keys, queries and value vectors of attention heads to employ representational superposition to represent more features than there are dimensions.

In contrast, neuron superposition is an instantiation of **computational superposition**, where a model can extract or compute with more features than it has dimensions (in this case neurons). If there exist more features than neurons that must be potentially computed, then each individual neuron must (on average) be involved in the computation of many features. Prior work on superposition(Elhage et al., 2022b) predominantly (but not entirely) focused on linear superposition, and we understand neuron superposition less well. As another example of computational superposition, preliminary findings suggest attention layers also employ superposition between heads to implement a larger number of skip-trigram circuits than would naively be possible (Jermyn et al., 2023).

Conceptually, computational superposition is a more complex phenomena. To perform superposition, the model must trade-off the benefit of representing more features against the costs of interference. With representational superposition the interference between features is just given by their dot product, and by embedding features as almost orthogonal vectors we can achieve low interference while representing many more features. But computational superposition involves nonlinearities (elementwise GELUs in the neuron case), which introduce significantly more interference in ways we do not fully understand.

### A.10 How can you conclude you found superposition as opposed to simply non-basis aligned features?

We think that our findings of superposition rest on two sub-claims: (1) that there exist more features than neurons to represent them and (2) that the neuron basis is meaningful, such that models are incentivized to align features with neurons *if* there are fewer features than neurons, and that superposition is implemented as a sparse combination of neurons.

We think claim (1) is evidently true, given the extremely long tail of features in internet text in addition to all the possible meaningful *n*-grams and other repeated patterns that model would want to memorize. In practice, models can detect and respond appropriately to a wide array of compound words and facts, seemingly far more so than the size of their vocabulary or number of neurons. As an example, just take the case of names of people. There are likely more people that GPT-3 knows about than it has neurons (5 million). A first and last name gets split into at least two tokens, but with most surnames being tokenized further. We think it would be impossible to fit all of the possible features of these people as just linear combinations of the token embeddings, therefore requiring some nonlinearity to detect the presence of a sequence of tokens corresponding to a name, and adding back the appropriate information about this person.

To further support claim (2), we run an experiment on the compound words dataset comparing the classification performance of the top 50 individual neurons in a layer to the top 50 dimensions in a random basis. In particular, for both the standard neuron basis, and the activation dataset when multiplied by a random $d \times d$ Gaussian matrix, we train 1-sparse probes for the 100 neurons with the largest class mean difference, and report the out of sample F1-score for the top 50 (averaged over 5 such random bases). Our results

in Figure 13 demonstrate that individual neurons are more predictive than random linear combinations of neurons, which implies that the neuron basis is meaningful.

Although there are usually 1-5 neurons with notably higher F1 score than the rest, the full top 50 neurons still maintain high F1 scores. This seems consistent with results from Section 8 of (Elhage et al., 2022b), where there exists a main neuron per feature, with a longer tail of less important neurons. Intuitively, just achieving linear separability is unlikely to be sufficient for overcoming interference. To minimize interference models likely want to maximize the separation between potentially interfering features, implying that the potential number of neurons in superposition may be much higher than the minimal number required to achieve near perfect classification performance. If the largest activation on negative examples is close to the smallest activation on positive examples, then using the represented feature for downstream computation will introduce significant noise. For a rough approximation, in Figure 14, we measure the logistic test loss, rather than F1 score, of $k$-sparse probes for a range of values of $k$. Again, we see that probes trained in the neuron basis generally have lower loss than those trained in a random basis. Perhaps more interestingly, there appears to be two regimes governing the loss with respect to sparsity, one with power law scaling, and then a kink with no returns to using additional neurons. It is plausible this kink gives an indication of the "true" sparsity used to implement features in superposition.

Finally, we note that these results likely understate the basis alignment, because they are with respect to an "easy" classification task: just distinguishing bigrams `XY` from bigrams `XZ` or `WY` rather than all possible $n$-grams, and that there likely exist many confounding contextual correlates that improve the performance of the random basis dimensions. Designing more careful experiments to untangle these different effects is an important area for future work.

### A.11 Should we ever expect to find monosemantic neurons?

Monosemantic neurons have been convincingly demonstrated in image classification models (Cammarata et al., 2021), but we are not aware of work rigorously showing monosemanticity of transformer language model neurons. So on an empirical front, this is an open question. It has been shown in toy models (Elhage et al., 2022b) that in the presence of significant differences in feature importance (i.e., expected reduction in loss from representing that feature) and for continuous features that the most important features will have dedicated monosemantic neurons. Textual features vary significantly in importance (e.g., detecting Python code is far more important than knowing some niche fact) suggesting that crucial features may get their own neuron (e.g. our context neurons). However, for important binary features this is less clear. Further, even important and highly prevalent features likely have rare features that they are anti-correlated with, and may be represented in superposition with these. Rare features are a case that may be particularly hard to detect.

### A.12 Which sparse probing method should I use?

It depends. If your experiments must be extremely fast or scalable, then simply doing maximum mean difference is likely best. If you are less runtime sensitive and want to sweep over a large range of $k$, then adaptive thresholding is most appropriate. Finally, if you require any sort of formal guarantees then you should use optimal sparse probing, with the caveat that this requires potentially substantial compute time.

### A.13 Why think of features as directions?

This is known as a linear, decomposable representation (Elhage et al., 2022b)—the model's activations may be decomposed into independently varying features, and these features correspond to directions in activation space. The intuition behind this is that the primary capability of models is doing linear algebra—addition and matrix multiplication, which further breaks down into addition, scalar multiplication, and projecting onto specific directions. Given these capabilities, it is especially natural for a model to represent features as directions: if a later layer wants to access a feature it can project onto that feature's direction, a neuron can easily access and combine multiple features, features can vary independently, and the component in that feature direction represents the strength of that feature. If a model used some more complex and non-linear representation then it would need to dedicate downstream parameters to decoding these, a costly

endeavour. Previous findings of monosemantic neurons in image models(Cammarata et al., 2021) and reverse-engineering toy models on mathematical tasks(Nanda et al., 2023; Chughtai et al., 2023) have found linear features, however the empirical evidence base here is slim. Mechanistic interpretability is still a young and pre-paradigmatic field, and we see gaining empirical evidence for foundational assumptions like this as a key area of future work. If the features-as-directions hypothesis were false, the task of reverse-engineering models would seem far more daunting: models are extremely high dimensional objects, and we must be able to decompose them *somehow* into independently meaningful units in order to evade the curse of dimensionality and understand them.

A naive way to test this hypothesis is by using linear and non-linear probes (e.g., a one hidden layer MLP) to extract features: if non-linear probes work and linear probes do not, then this is evidence that those features are represented non-linearly. However, this can be illusory, as the model may instead linearly represent simpler features and the non-linear probe itself does the computation to find the more complex feature. In the extreme, we could imagine having our 'probe' be GPT-3 trained on the token embeddings—this can surely probe for any feature, represented or not! This is a concern even with linear probes, eg a model may represent whether a shape is red and whether it is a triangle as directions in space, and a linear probe on the sum of these directions may seem to indicate an 'is a red triangle' feature.

There was a recent natural experiment in the literature supporting the features as direction hypothesis. (Li et al., 2022) trained a model to predict the next move in the board game Othello, and were able to probe for an emergent representation of the board state. They could only find this representation with non-linear probes, not linear probes, yet could validate this representation with causal interventions. However, follow-up work (Nanda, 2023b) showed that there was in fact a linear representation of the board state, but that as the model predicted both black and white moves, it was in terms of whether a cell had color equal to the current player or to the current opponent. We take results like this as promising additional evidence that the features-as-directions hypothesis has real predictive power.

### A.14 What are possible conceptual models for what MLP layers are actually doing?

Perhaps the main model, as hypothesized and studied in (Geva et al., 2020), is that of a key-value store (note that the key and value terms here are unrelated to those used in the attention layers). A key-value store can be used in tasks such as memorization and factual recall, where the model wants to look up a fact, such as the location of the Eiffel Tower. If the residual stream contains both the features "Eiffel" and "Tower," then a neuron's key could be the sum of both of those features, and the value could be the feature indicating that the current token is in Paris.

The non-linearity of the MLP might serve several possible roles, such as thresholding, where only if the "Eiffel" and "Tower" features are present, do we conclude that it is in Paris. It may also perform **saturation**: for features where the model must accumulate many small pieces of evidence but the underlying ground truth is bimodal, such as "is Python code", the model may want to have the same thresholding . Additionally, some form of translation or memory management could be necessary. If features like "is in Paris" are only used in middle layers, it would not be in the model's interests to produce the "is in Paris" feature earlier on for fear of causing interference. Another possible role is that of activation range management. More broadly, neurons seem well-suited to representing Boolean functions such as AND, which is inherent in studying simple structures such as bigrams and trigrams. An AND can be implemented by stating that if the features A and B are both present with size 1, then only if their sum is present with size 2 does the neuron fire, and otherwise it is set to zero (e.g., $x, y \rightarrow \text{ReLU}(x + y - 1)$).

Another role that neurons might play is that of disambiguation. There are certain tokens which occur in very different contexts, but where the same token arises, such as "Die" in English corresponding to "death," "dice," or "Die" as a common token in languages such as German or Dutch or Afrikaans. Neurons that disambiguate "Die" have been observed in the literature (Elhage et al., 2022a; Coenen et al., 2019). In some sense, this can be thought of as having a more general "this is in Dutch" neuron producing an "is in Dutch" feature, and having the neuron having the "Die in Dutch" feature be the linear combination of the "is in Dutch" feature and the "Die" token is present feature. Depending on perspective, this could be considered a superposed, or distributed representation of the "Die in Dutch" feature.

One useful model for early layer MLPs is **detokenization** (Elhage et al., 2022a): taking the raw input of tokens and converting it to more useful semantic concepts, a la the *n*-gram detectors we study. These are analogous to sensory neurons in biological systems, or gabor filters and high-low frequency detectors of early vision layers (Mehrotra et al., 1992; Olah et al., 2020a): neurons that take the raw sensory input and convert it to a more useful form. A non-obvious consequence of this is that detokenization neurons may *simultaneously* serve as key-value stores. An "Eiffel Tower" *n*-gram detector may have an "is in Paris" feature that it outputs, in addition to the "is Eiffel Tower" feature. From another perspective, assuming the representations are linear, it could be the case that the "is in Paris" feature is nothing more than the average of features like "is Eiffel Tower", "is the Louvre", etc.

The converse model for late layer MLPs is **retokenization** (Elhage et al., 2022a): converting the features represented in some semantic space into the concrete output tokens, analogous to motor neurons. This is likely to be particularly important with multi-token words: if the model completes a prompt like "A famous landmark in Paris is" with " Eiffel Tower", it must in fact output 4 separate tokens (" E","iff","el"," Tower"), and this is natural to be performed by late layer MLPs on the current and subsequent tokens. Though, a notable difficulty in detecting these neurons is distinguishing between neurons converting an "output Eiffel Tower" feature on the " is" token to concrete outputs, from neurons that detect and continue the *n*-gram of " Eiffel Tower", but only figure it out after the " E" and "iff". This is a similar to difficulties in evaluating model performance when predicting multi-token outputs.

One interesting application of neurons in transformers that does not occur with simpler models such as convolutional networks is neurons engaging with and enhancing the function of attention heads. Models with a single attention-only layer can exhibit functions such as skip trigrams, but where this has bugs (Elhage et al., 2021), a neuron could be used to fix this. They could also compose with important attention head circuits, such as induction heads, to help clarify edge cases. Induction heads detect and continue repeated text, but if a token arises in multiple different contexts in the prior text with different subsequent tokens, we may want the induction heads to not activate at all. This may be easier with clarifying neurons than with an attention-focused circuit.

Many of our intuitions here involving thinking of GELU as essentially the same activation as a ReLU, while we know that GELUs perform better in practic e(Hendrycks & Gimpel, 2016). Though this may be down to ease of optimisation, we speculate that the "bowl" of the GELU, in particular the fact that it is close to a quadratic when close to the original, may allow the model to express more complex non-linear functions. We would be particularly excited to see future work exhibiting such case studies.

### A.15 Did you train your probes on pre-GELU or post-GELU activations?

We train our probes on the post-GELU neuron activations (i.e., $\text{GELU}(W_{in}x)$) because there's a linear map from these to residual stream, so linear combinations of them are meaningful to the network, in a way that pre-GELU linear combinations are not. We often choose to plot histograms of pre-GELU activations for individual neurons for clarity, to better show the negative range, and to avoid an enormous spike around zero from all highly negative activations.

### A.16 Did you have any negative results?

Yes. We had two types of feature datasets which did not appear to be sparsely represented: falsehoods in the counterfact dataset or the occurrence of particular suffixes and prefixes, though both attained fairly high accuracy with a dense probe. For prefixes and suffixes, since these are purely a property of single tokens, we believe these aren't likely to get dedicated neuron representation since the feature already exists within the token embedding.

For counterfact, a dataset of true and false factual completions, we hypothesized there might exist some dedicated "falsehood" neuron. However, sparse probes performed quite poorly, while increasing $k$ continued to yield performance gains until about $k = 1024$ where we achieved 90% accuracy on the best layer. Our interpretation of this is that the probe is picking up some pattern of confusion or dissonance, associated with an unexpected completion, rather than a true falsehood or truth feature.

Our last negative result was in attempting to causally implicate the factual neurons. That is, run ablations that degrade the performance of few shot prompting based classification of people's gender, occupation, and whether they are alive or not. However, ablating individual neurons seemed to have very limited effect, indicating that either the "fact" was computed previously, and these neurons are simply responding to this fact, or that there is some amount of ensembling or redundancy (Dalvi et al., 2020) that is robust to the deletion of a single neuron.

### A.17  What do you mean by the model "wants"?

Throughout this FAQ, we sometimes use anthropomorphic language, such as that the model "wants" to minimise interference. We do not, of course, think that these small models are capable of having wants or intentionality in the sense of a human! In fact, it would be more accurate to say that stochastic gradient descent selects for models with low expected loss, and that model weights that minimise interference seem likely to have lower expected loss. But we believe that notions like "want" can be valuable intuition pumps to reason about what algorithms may be expressed in a model's weights, so long as the reader is careful to keep in mind the limits and potential illusions of anthropomorphism.

## B  Experimental Details

### B.1  Models

We study the Pythia model suited (Biderman et al., 2023) which was trained on The Pile (Gao et al., 2020), with each model trained for the same number of steps with an identical data ordering. These models follow a fairly standard architecture, but use parallel attention and rotary positional embeddings. See Table 1 for the architectural parameters of each model studied.

| Model | $n_{\text{layers}}$ | $d_{\text{model}}$ | $n_{\text{heads}}$ | $d_{\text{head}}$ | Total Number of Neurons |
|---|---|---|---|---|---|
| Pythia 70M | 6 | 512 | 8 | 64 | 12288 |
| Pythia 160M | 12 | 768 | 12 | 64 | 36864 |
| Pythia 410M | 24 | 1024 | 16 | 64 | 98304 |
| Pythia 1B | 16 | 2048 | 8 | 256 | 131072 |
| Pythia 1.4B | 24 | 2048 | 16 | 128 | 196608 |
| Pythia 2.8B | 32 | 2560 | 32 | 80 | 327680 |
| Pythia 6.9B | 32 | 4096 | 32 | 128 | 524288 |

Table 1: Hyperparameters of Pythia models studied.

### B.2  Datasets

Table 2 contains summary statistics of all of our probing datasets in addition to the full list of features in each. We briefly describe the design, preprocessing, and motivation of each one.

**Natural Language**  We used both the raw text and the labels from the EuroParl subset of the pile, which contain a large number of parliamentary proceedings in many different languages. Many documents were quite a bit longer than our context length, so for each, we took a random contiguous sequence from the document. This choice was also made to minimize any context clues from the beginning of the document about the country speaking. For probing, we performed mean-aggregation. That is, computing the activation for all non-padding tokens in the sequence and using the average neuron activations as the representation for the sequence to be used for probing. One problem with this dataset is that all of the sequences are similarly styled, and about similar topics, hence our probes could be picking up on a particular style or topic feature rather than a true language feature.

| Dataset | Sequences | $n_{ctx}$ | Non-pad tokens | Source | Pos ratio | Total Features |
|---|---|---|---|---|---|---|
| part-of-speech | 1438 | 512 | 281044 | EWT | 0.20 | 16 |
| dependencies | 1438 | 512 | 281044 | EWT | 0.20 | 29 |
| morphology | 1438 | 512 | 281044 | EWT | 0.20 | 22 |
| code language | 5397 | 512 | 2757867 | pile-github | 0.11 | 9 |
| natural language | 28084 | 512 | 14350924 | pile-europarl | 0.11 | 9 |
| text features | 10000 | 256 | 2248714 | pile-test-all | 0.26 | 11 |
| data subset | 8413 | 512 | 4299043 | pile-test-all | 0.11 | 9 |
| compound words | 167959 | 24 | 4031016 | pile-test-all | 0.20 | 21 |
| latex | 4486 | 1024 | 4589178 | pile-arxiv | 0.26 | 14 |
| wikidata sex or gender | 6000 | 128 | 688085 | pile-test-all | 0.50 | 2 |
| wikidata is alive | 6000 | 128 | 688179 | pile-test-all | 0.50 | 2 |
| wikidata occupation | 6000 | 128 | 677512 | pile-test-all | 0.17 | 6 |
| wikidata athlete | 5000 | 128 | 575350 | pile-test-all | 0.20 | 5 |
| wikidata political party | 3000 | 128 | 337755 | pile-test-all | 0.50 | 2 |

Table 2: Summary statistics of probing datasets studied.

**Data Subset**   Similar to the above, we randomly selected several thousand sequences from each data subset of the Pile test set, where the labels were implied. We also used the same random sub-sequencing and mean aggregation.

**Programming Languages**   For our programming language dataset, we took all of the github subset from the Pile test set. We then used a code recognition package to classify the type of code, and only include the source files with a prediction of over 90% confidence. A challenge with code is that it includes many tokens which aren't code (licenses, copywrite, comments, etc.). As a very coarse cleaning attempt, we ignore the first 50 tokens in the sequence when applying mean aggregation, to avoid beginning of file boilerplate. However, for some languages like HTML or XML that likely contain substantial plain text, this does little to help (and hence we see the lowest accuracy on HTML and XML).

**Compound Words**   For our compound words dataset, we computed the top thousand alphabetical bigrams XY over the pile test set. We then filtered for bigrams where $P(X = x|Y = y)$ and $P(Y = y|X = x)$ were below 0.3 to make the prediction task less trivial. We then manually filtered to bigrams with the property that the first and second word together means something quite distinct from either word separately. For the probing dataset, we then included 2000 short (24 tokens) sequences from The pile ending with the bigram XY, 4000 examples ending with XW, W$\neq$Y, and 4000 examples ending with ZY, Z$\neq$X. We then probe on the activations of the last token of every sequence.

**Latex Features**   For our latex feature dataset, we included the first 1024 tokens of all documents from the ArXiV subset of the Pile test set which contained at least 1024 tokens. We mostly relied on regular expressions to extract features, though used a simple finite state machine to parse the math text. For each of the features involving any sort of "containment," (which is most of them), we used the policy that all tokens within the containment counted as being part of the feature, but any tokens defining the containment for not. For instance for `is_subscript`, the "_{" token is not included, but anything within the braces is. For our positive class, we randomly sample tokens from the set of all positive tokens. For features with a natural complement (e.g., `is_numerator` and `is_denominator`) we make the negative class be occupied by half of tokens within the complement, and half random tokens.

**Text Features**   For our plain-text feature dataset, we similarly extracted features using regular expressions, but just applied to the raw token strings. Therefore, for our text features, a feature could be correctly predicted simply by partitioning the tokens into two classes, with no additional context. We sampled tokens randomly from a subset of the Pile test set.

| Dataset | Features |
|---|---|
| part of speech | AUX, ADP, VERB, ADJ, X, CCONJ, PROPN, NOUN, INTJ, SYM, PRON, DET, SCONJ, ADV, PUNC, NUM |
| dependencies | aux:pass, acl:relcl, nsubj, xcomp, flat, cc, mark, acl, ccomp, appos, root, nmod:poss, aux, amod, nsubj:pass, obj, obl, det, advmod, punct, parataxis, conj, case, list, advcl, cop, compound, nummod, nmod |
| morphology | eos_True, Person_2, Gender_Fem, VerbForm_Inf, PronType_Dem, Gender_Masc, first_eos_True, Gender_Neut, VerbForm_Part, NumType_Card, PronType_Int, PronType_Prs, Person_3, Tense_Past, Number_Plur, PronType_Art, Voice_Pass, PronType_Rel, VerbForm_Ger, Mood_Imp, Person_1, VerbForm_Fin |
| code language | Python, XML, Java, C++, HTML, C, Go, PHP, JavaScript |
| natural language | Swedish, Portuguese, German, English, French, Spanish, Greek, Italian, Dutch |
| text features | leading_capital, no_leading_space_and_loweralpha, all_digits, is_not_ascii, has_leading_space, contains_all_whitespace, all_capitals, is_not_alphanumeric, contains_whitespace, contains_capital, contains_digit |
| data subset | github, pubmed_abstracts, stack_exchange, wikipedia, freelaw, hackernews, arxiv, enron, uspto |
| compound words | mental-health, magnetic-field, trial-court, control-group, human-rights, north-america, clinical-trials, high-school, third-party, public-health, cell-lines, living-room, second-derivative, credit-card, social-media, prime-factors, federal-government, social-security, blood-pressure, gene-expression, side-effects |
| latex | is_superscript, is_inline_math, is_title, is_subscript, is_reference, is_denominator, is_author, is_numerator, is_display_math, is_math, is_abstract, is_frac |
| wikidata sex or gender | is_female, is_male |
| wikidata is alive | true, false |
| wikidata occupation | is_actor, is_athlete, is_journalist, is_politician, is_researcher, is_singer |
| wikidata athlete | is_american_football_player, is_association_football_player, is_baseball_player, is_basketball_player, is_ice_hockey_player |
| wikidata political party | is_democratic_party, is_republican_party |

Table 3: All feature within the feature collections.

**Linguistic Features**   For our linguistic features, we use the text and labels from the well known Penn Treebank Corpus. The dataset is provided at a sentence level, so we first merge sentences from the same documents, and then perform a token alignment and character mapping to adapt the dataset labels to the specific tokenization of the Pythia models. For features that apply to all tokens (like part-of-speech or dependency relations), we have no restrictions on the tokens sampled for the negative class. For features restricted to tokens of specific types (e.g. verb tense), we restrict the negative examples to be from the same token class (e.g. only verbs).

**Wikidata Features**   To create a Wikidata feature dataset for a specific property (e.g. gender, occupation), we search through a JSON dump of all Wikidata entities and compile a table containing the names and relevant property value for each person with data on the chosen property. When there are multiple Wikidata entities with the same name, we choose the entity with more total Wikidata properties defined as a heuristic for relevance. We then search the text of the test set of The Pile for instances of these names and filter out all examples with names that occur only once in the dataset. We tokenize each matching string and truncate to 128 tokens such that the tokenized name is at the end (if there are sufficient preceding tokens, otherwise we pad the tokenized string to length 128). We further filter the tokenized examples to ensure that each name occurs no more than three times in each feature dataset. This helps us find general knowledge neurons that

aren't overly impacted by individual people. Then when probing, we consider only the activations for the tokens of the surname for each example to reduce the effect of correlations between first names and certain properties (such as gender).

### B.3 Feature Selection Methods

Throughout, assume we have a dataset of token activations for $n$ tokens and $d$ neurons $X \in \mathbb{R}^{n \times d}$, and a vector of labels $y = \{-1, 1\}^n$. We refer to the set of tokens in the positive class $P$ and the set of tokens in the negative class $N$. Below we describe the different feature selection methods in more detail. With the exception of optimal sparse probing and adaptive thresholding, we can think of these approaches as scoring algorithms that can be used to rank the different neurons by importance. Then, to get a $k$ sparse classifier, in all cases one just selects the $k$ neurons with highest score $s$.

**Mean Difference**   The score is given by the average mean difference between classes for each neuron. That is, for neuron $j$ we have $s_j = \frac{1}{|P|} \sum_{i \in P} X_{ij} - \frac{1}{|N|} \sum_{i \in N} X_{ij}$.

**Mutual Information**   We use the algorithm presented by (Ross, 2014), to compute the mutual information between continuous data (in this case each neuron independently) and a discrete target relying on a nearest-neighbor approximation. The score for each neuron is then just the estimated mutual information.

$L_1$ **regularized**   We train a dense logistic regression probe on the activations with $L_1$ regularization. The score for each neuron is the absolute value of the corresponding coefficient of the dense probe.

**F-statistic**   We utilize a one-way ANOVA test to evaluate the relationship between the continuous data of each neuron independently and a discrete target. The F-statistic is calculated by partitioning the total variability into between-group and within-group components, represented by sum of squares between groups (SSB) and sum of squares within groups (SSW), respectively. The ratio of the mean square between groups (MSB) to the mean square within groups (MSW) is computed as the F-statistic. A higher F-statistic indicates a larger discrepancy between group means and is used as the score for each neuron.

**Optimal Sparse Probing**   We train a cardinality-constrained support vector machine (SVM) with hinge-loss trained to provable optimality using the cutting planes technique as described in (Bertsimas et al., 2021). The selected coefficients are simply those selected by the classifier.

**Adaptive Thresholding**   Starting from $k = d_{mlp}$ (or some large $k$ on a filtered set of neurons from a different heuristic), we train a series of logistic regression probes with elasticnet regularization (that is combined $l_1$ and $l_2$ regularization). Given a schedule for reducing $k$, at each step $t$, we take the $k_t < k_{t-1}$ neurons with highest absolute coefficient magnitude and retrain a logistic regression probe on these neurons. This process enables us to sweep through many values of $k$ while leveraging past computation to achieve better feature selection.

### B.4 Experimental Procedure

Here we provide additional details on the experimental procedure regarding both the feature selection experiments and the sparsity with scale experiments within Section 4.

Perhaps most significantly, for all experiments we preceded probe training with a heuristic filtering step, where we only trained probes on the top 1024 neurons, as judged by mean difference, for each combination of layer, feature, and model. This was done for computational reasons, as computing the mutual information, training dense probes, and optimal sparse probing methods are very expensive to run on hundreds of features for many layers containing in excess of 10,000 neurons. For optimal sparse probing, we only used the top 50, and set a timeout of 60 seconds.

All classifiers were trained with balanced class weights to account for class imbalances in the dataset. Finally, we selected hyperparameters using a small subset of feature and model combinations, and then used the best

| Method | Sparsity | | | | | | | | Runtime (s) |
|---|---|---|---|---|---|---|---|---|---|
| | 1 | 2 | 3 | 4 | 5 | 6 | 7 | 8 | |
| **Random** | 0.573 | 0.629 | 0.663 | 0.690 | 0.712 | 0.729 | 0.743 | 0.755 | 0.257 |
| **MMD** | **0.832** | 0.857 | 0.867 | 0.874 | 0.880 | 0.884 | 0.887 | 0.890 | 0.292 |
| **FS** | **0.832** | 0.856 | 0.867 | 0.874 | 0.880 | 0.884 | 0.887 | 0.890 | 0.303 |
| **LR** | 0.818 | 0.853 | 0.871 | 0.882 | 0.890 | **0.897** | **0.902** | **0.906** | 7.796 |
| **MI** | 0.831 | 0.851 | 0.863 | 0.869 | 0.876 | 0.880 | 0.884 | 0.886 | 32.241 |
| **OSP** | 0.831 | **0.861** | **0.876** | **0.884** | **0.891** | 0.896 | 0.899 | 0.902 | 53.028 |

Table 4: Comparison of sparse feature selection methods. Results are for the out-of-sample F1 score averaged for the best scoring layer for each combination of model and feature.

performing hyperparameters for all other experiments, as doing seperate tuning for every trial would be too computationally expensive.

### B.5    Feature Selection Results

For each combination of model, feature, and layer described above, we train a sparse probe for values of $k = 1, \ldots, 8$ using the features selected by the aforementioned subset selection methods. In Table 4, we report the out-of-sample F1 score of each method averaged across models and features, and for the maximum across layers since features tend to be only represented in specific layers.

While our results are averaging over many trials, several points stand out. The top methods are always within 1% of each other, with no method strictly dominating the others. Although OSP comes with optimality guarantees, we set a one minute timeout, which is frequently reached especially for larger values of $k$. Moreover, this provable optimality is with respect to the specific level of $l_2$ regularization which often needs to be turned fairly high to speed up convergence (Bertsimas et al., 2021). Nevertheless, it is impressive that such fast and simple heuristics as the mean difference are comparable to far more sophisticated approaches, with even random neurons containing substantial information of the target task. This suggests that perhaps the best design is a two-stage approach where very fast heuristics are used to find relevant neurons and layers, and optimal methods are used to improve and verify the solution.

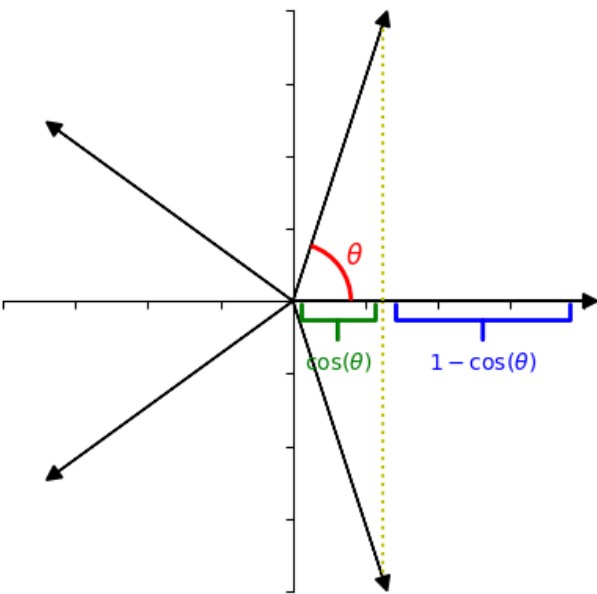

Figure 6: Intuition for representing 5 features in 2 dimensions in superposition.

## C    Superposition Construction

Consider a set of $n$ binary and mutually exclusive features (i.e., one-hot vectors like $n$-grams), that we want to embed in two dimensions, and later losslessly recover. Specifically, given a one hot vector $x \in \mathbb{R}^n$, we seek a $W \in \mathbb{R}^{n \times 2}$ such that

$$x = \text{ReLU}(WW^T x + b)$$

where $\text{ReLU}(x) = \max(x, 0)$. By embedding each point to be equally spaced around a circle of radius $\alpha$ (i.e., $w_i = \alpha \cos(2\pi i/n), \alpha \sin(2\pi i/n)]$ , we get that

$$(WW^T e_i)_j = \alpha^2 \cos\left(2\pi \frac{|i-j|}{n}\right)$$

where $\alpha = \|w_i\|_2$. To recover the initial one-hot representation, we require

$$\alpha^2 \left(1 - \cos\left(\frac{2\pi}{n}\right)\right) = 1 \quad \text{and} \quad b = -\alpha^2 \cos\left(\frac{2\pi}{n}\right)$$

Analyzing our proxy metric, $b\|w\|_2 = \frac{\cos(2\pi/n)}{(\cos(2\pi/n)-1)}$ monotonically decreases for $n$ with $n > 2$.

See (Elhage et al., 2022b) for a far more extensive set of experiments and results on related topics.

Note that this construction refers to residual stream superposition and *not* neuron superposition since we are assuming we have exactly as many neurons as features. However, we expect the same basic motif to clean up interference to hold for the case with fewer neurons than features. Moreover, it is plausible increased levels of residual stream superposition is associated with increased levels of neuron superposition. Hence we believe this construction, and therefore the presence of large weight norms and negative biases, to be suggestive evidence of neuron superposition.

Of course, it is possible factors other than superposition explain this observed weight pattern. Additionally, there might be multiple mechanisms for computing features in superposition (likely those that are not as binary or mutually exclusive as $n$-grams) that do not strongly affect the distributions of weights and biases. We would be very excited about future work that more carefully explained the source of these weight statistics, or that studied other concrete mechanisms or motifs enabling effective computation in superposition.

# D    Additional Results

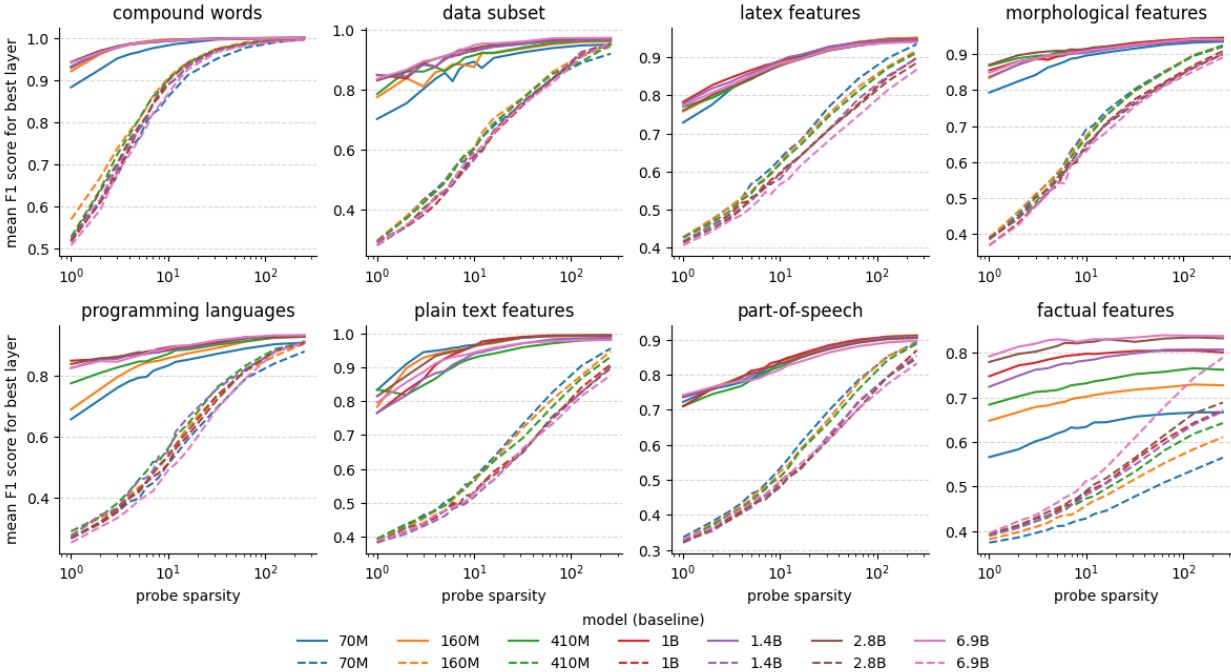

Figure 7: Difference between a $k$-sparse classifier using sparse features selection versus a random baseline. Specifically, the F1 score of training a classifier with $k$ coefficients on the full $n \times d$ activation dataset when multiplied by a random $d \times k$ orthogonal matrix.

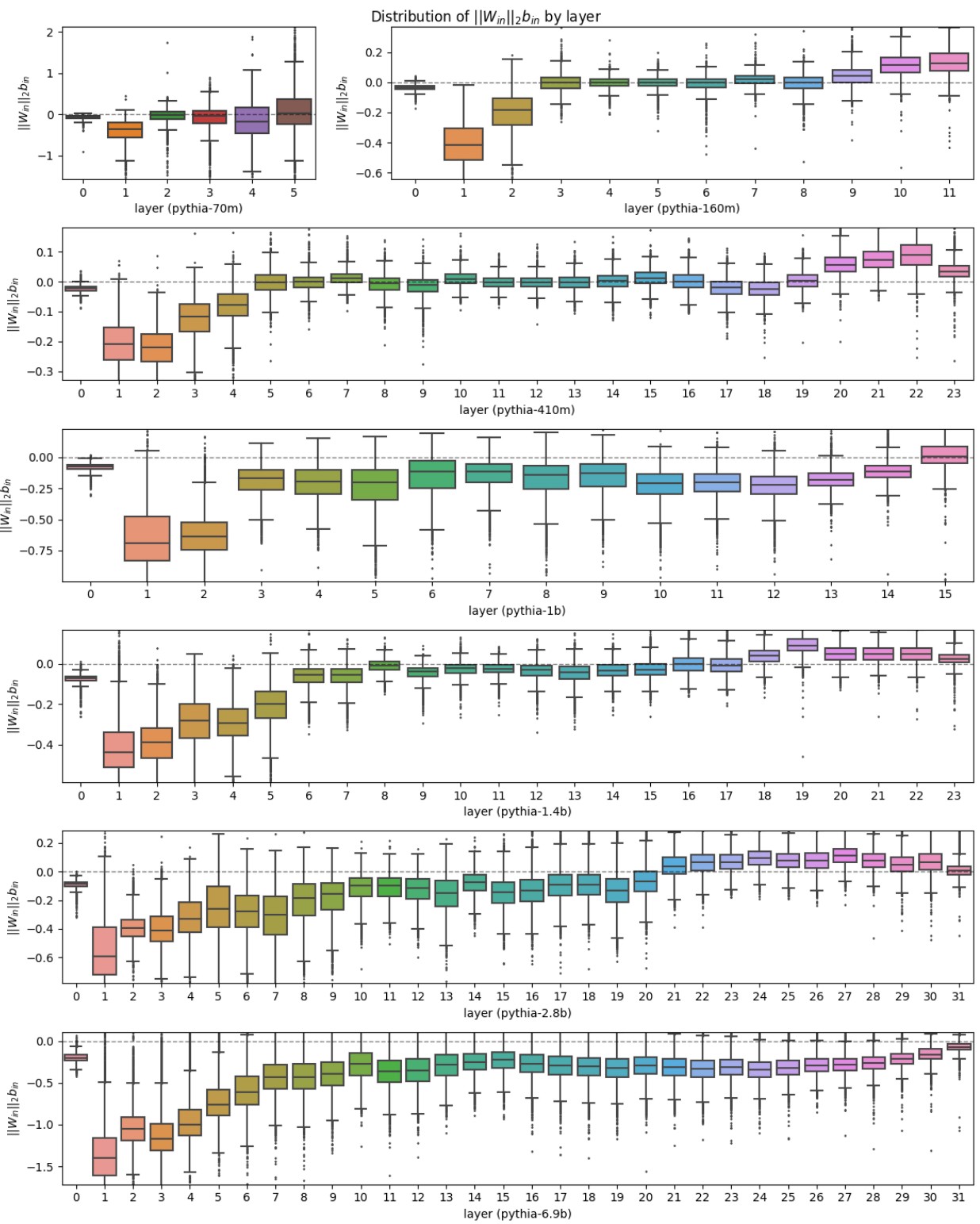

Figure 8: Distribution of input weight norms and biases for all Pythia models studied.

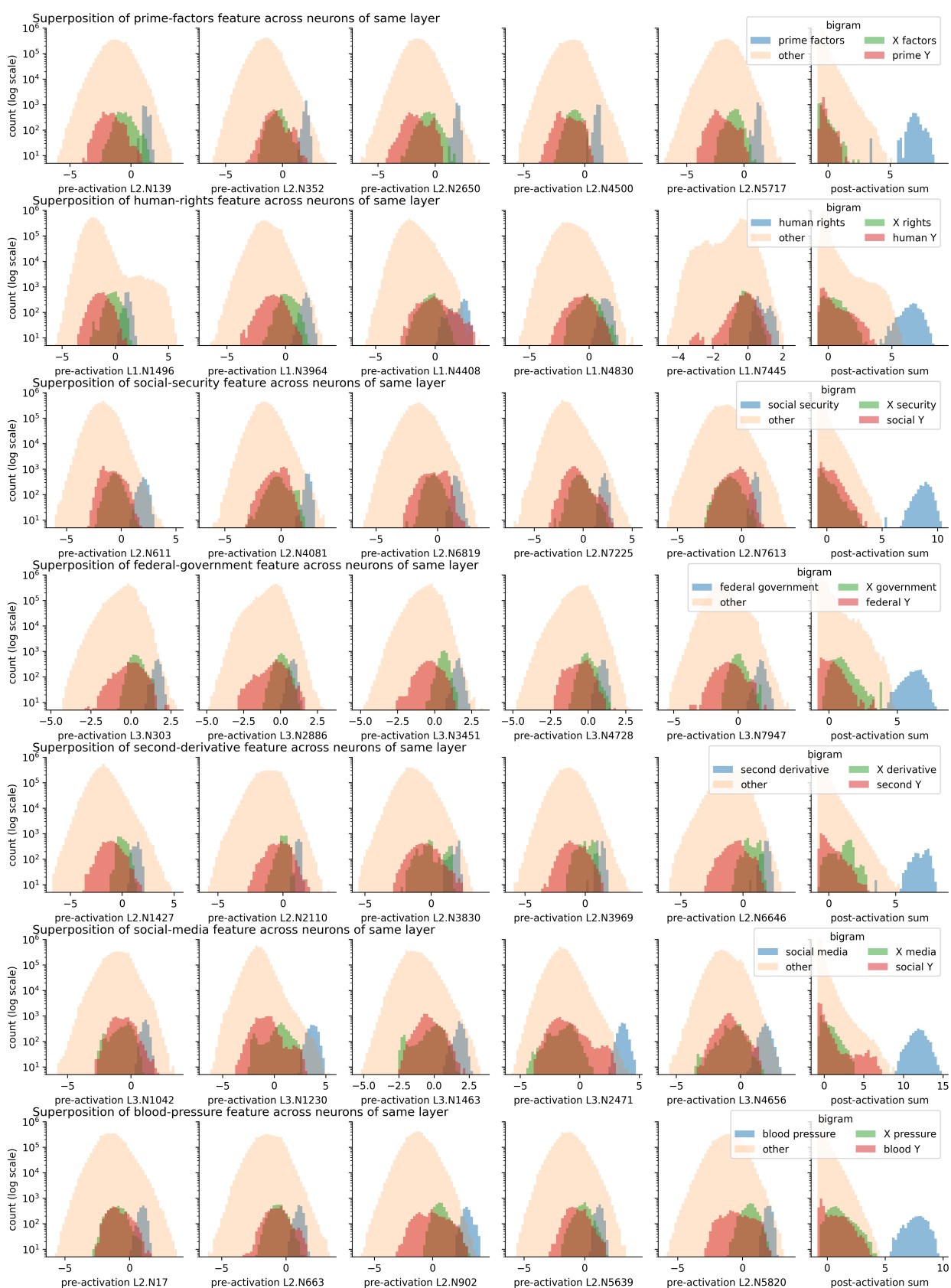

Figure 9: More examples of superposition of compound words in Pythia-1B

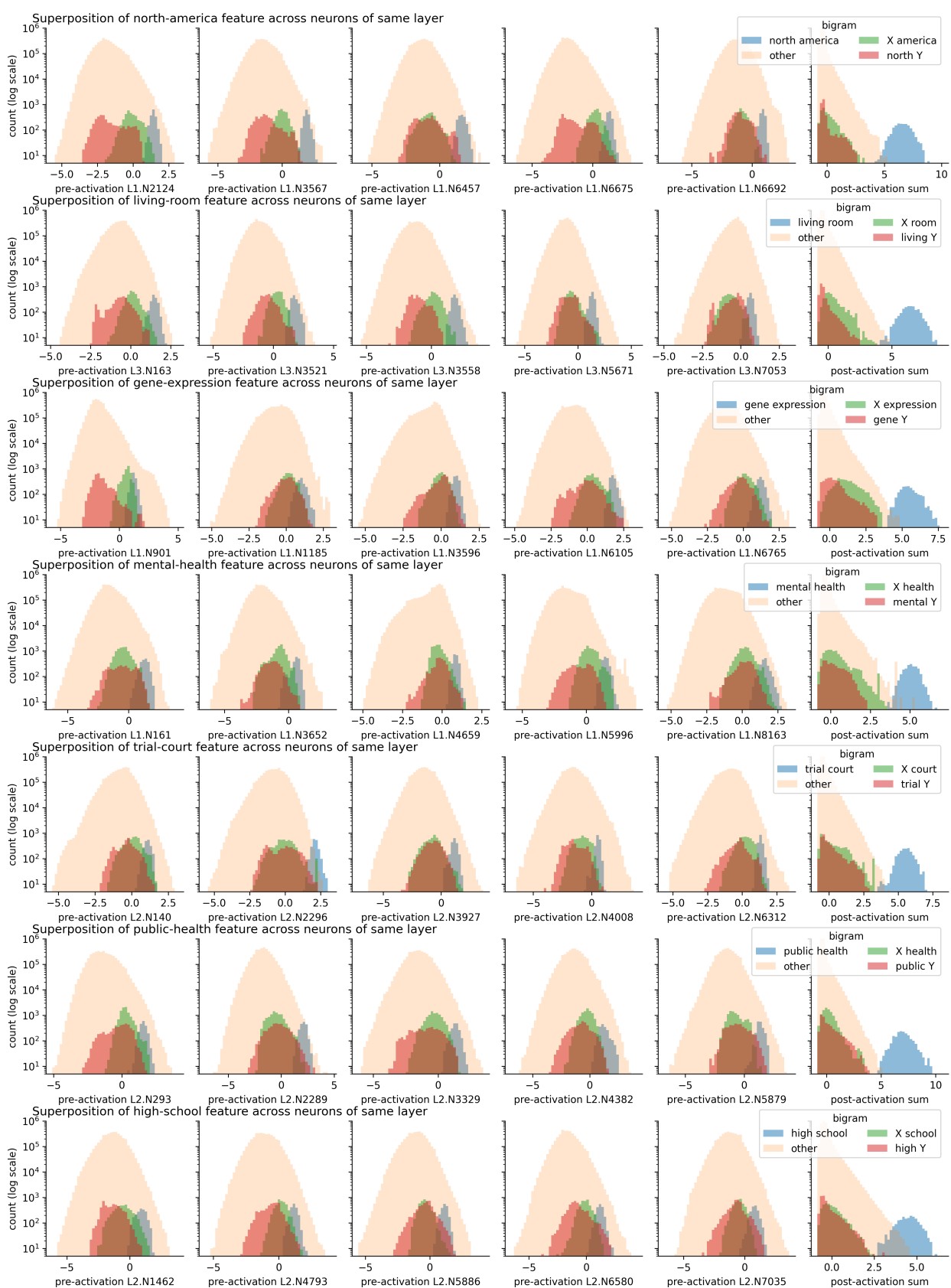

Figure 10: More examples of superposition of compound words in Pythia-1B

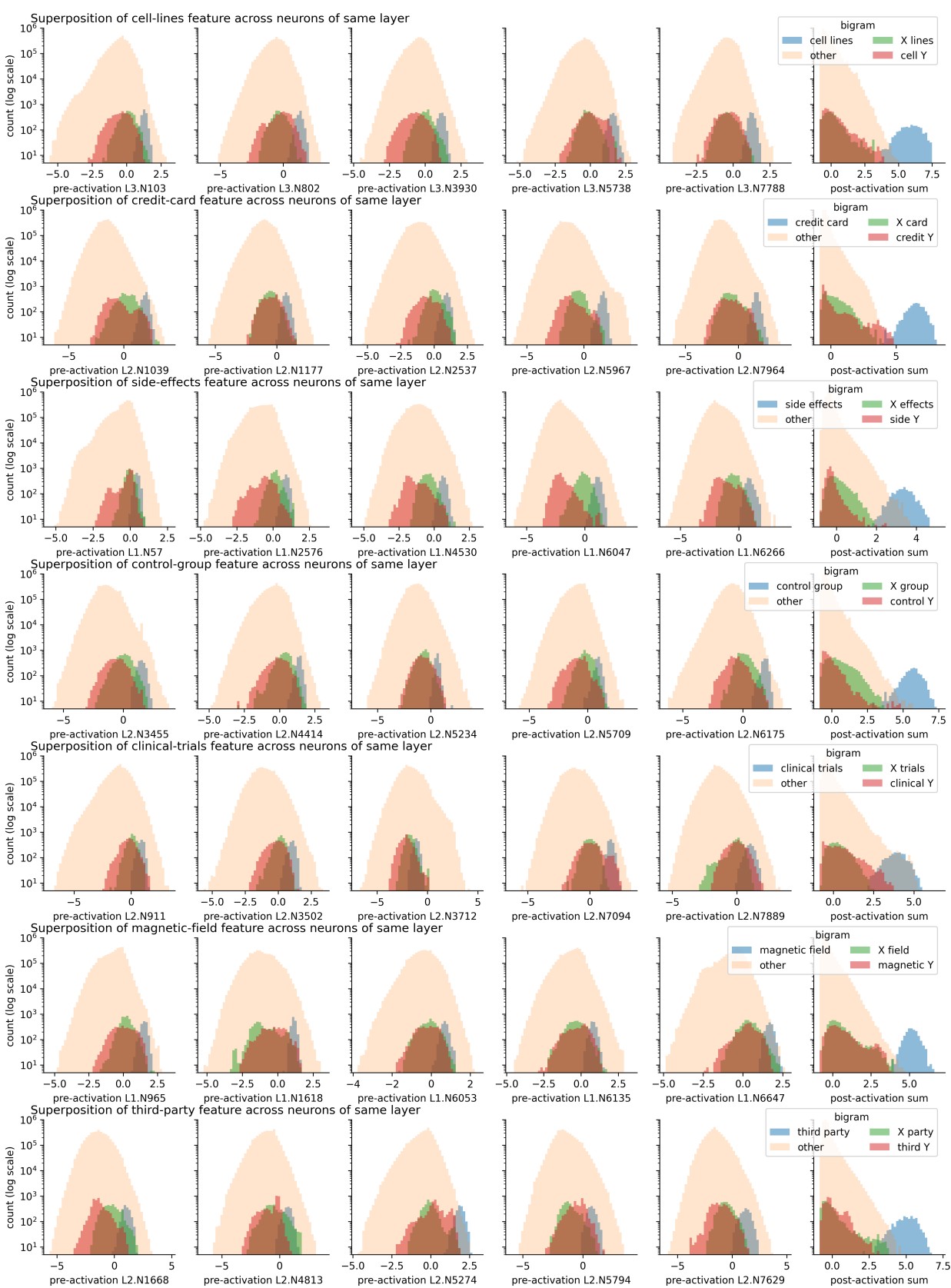

Figure 11: More examples of superposition of compound words in Pythia-1B

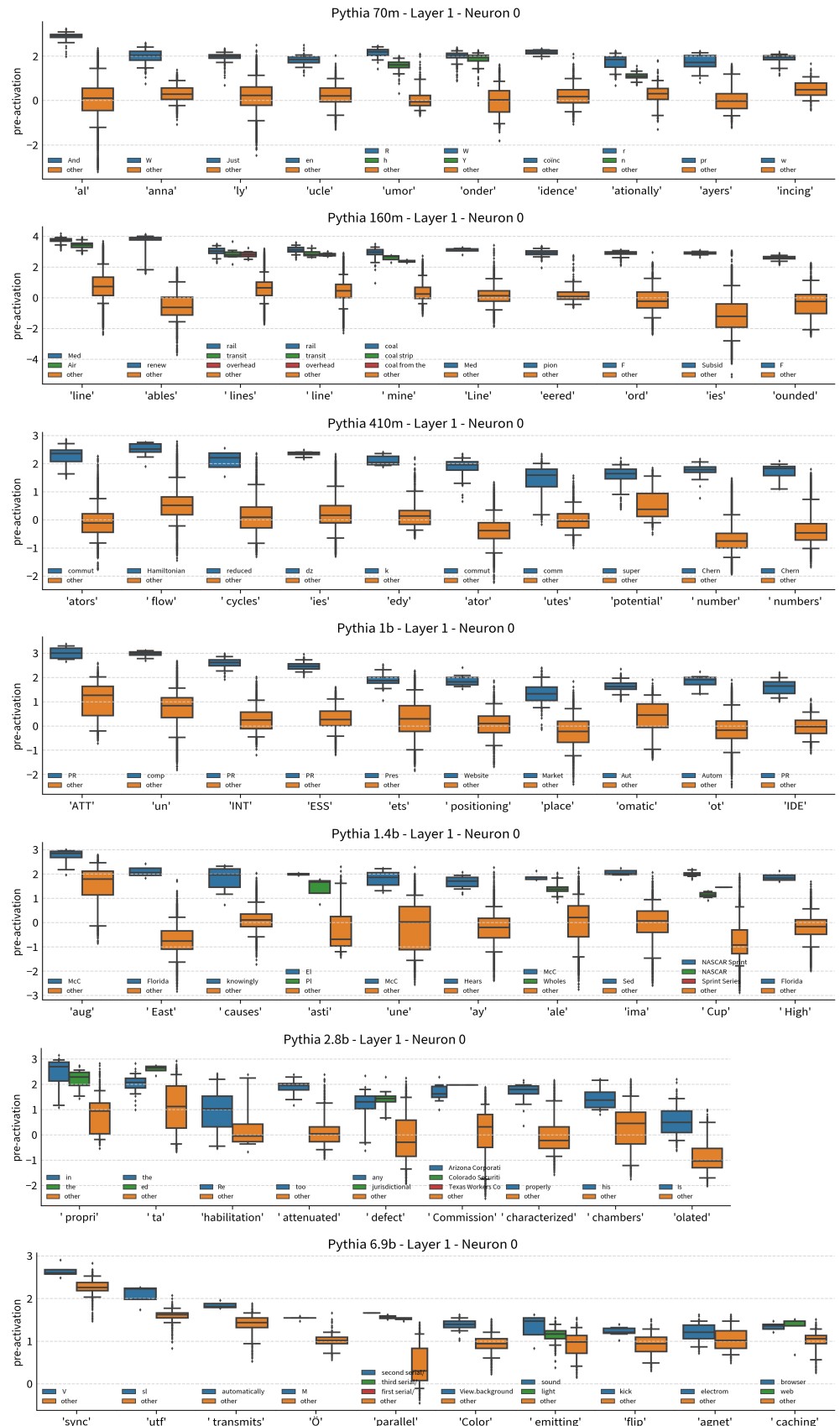

Figure 12: More examples of $n$-gram polysemanticity in layer 1 neuron 0 of all Pythia models studied.

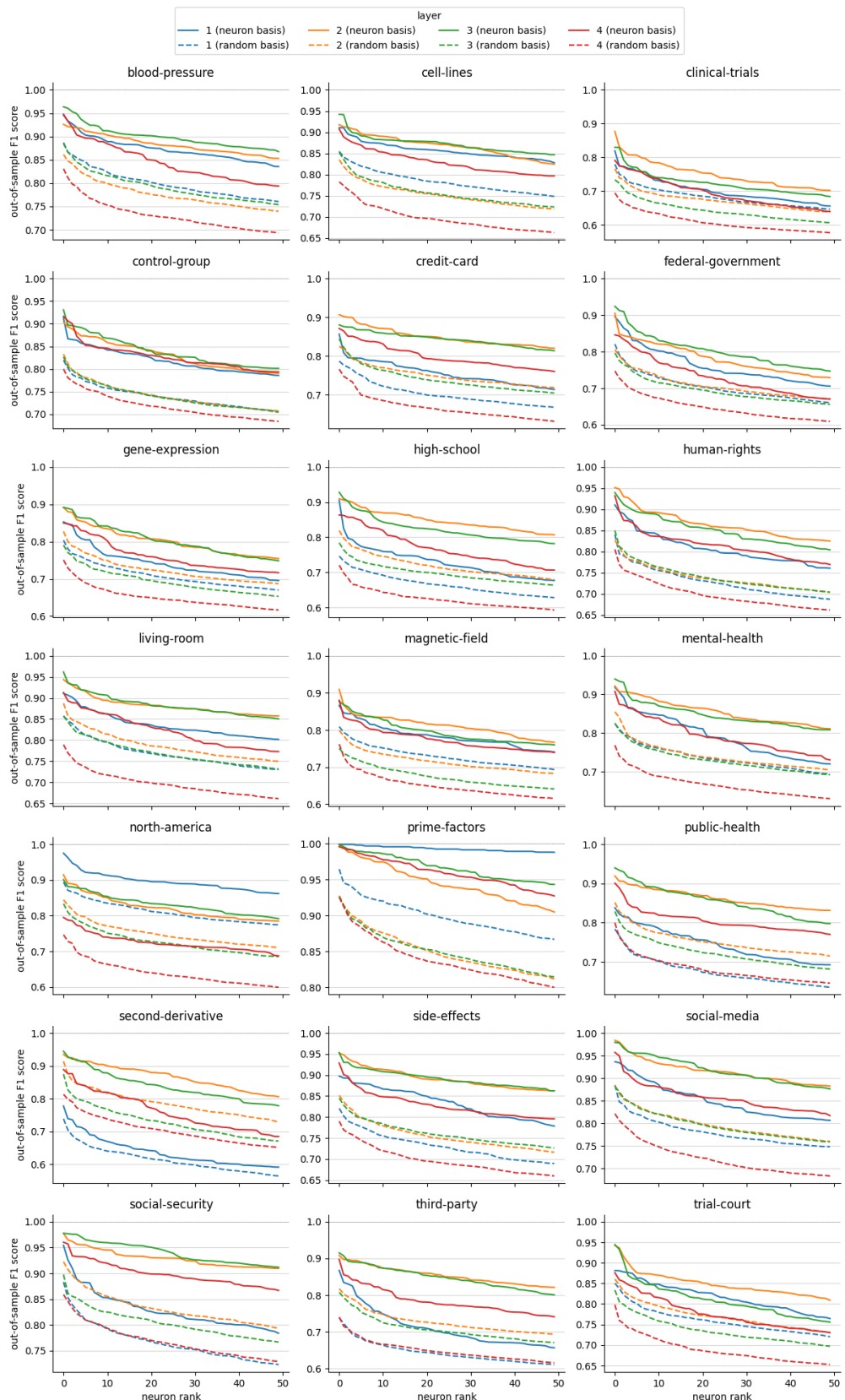

Figure 13: Demonstration of neuron basis alignment in early layers of Pythia-1B. Compares the compound-word classification performance of the top 50 neurons for layers 1-4 in the standard neuron basis, as opposed to a random basis (the post-activation dataset multiplied by a random Gaussian $d \times d$ matrix).

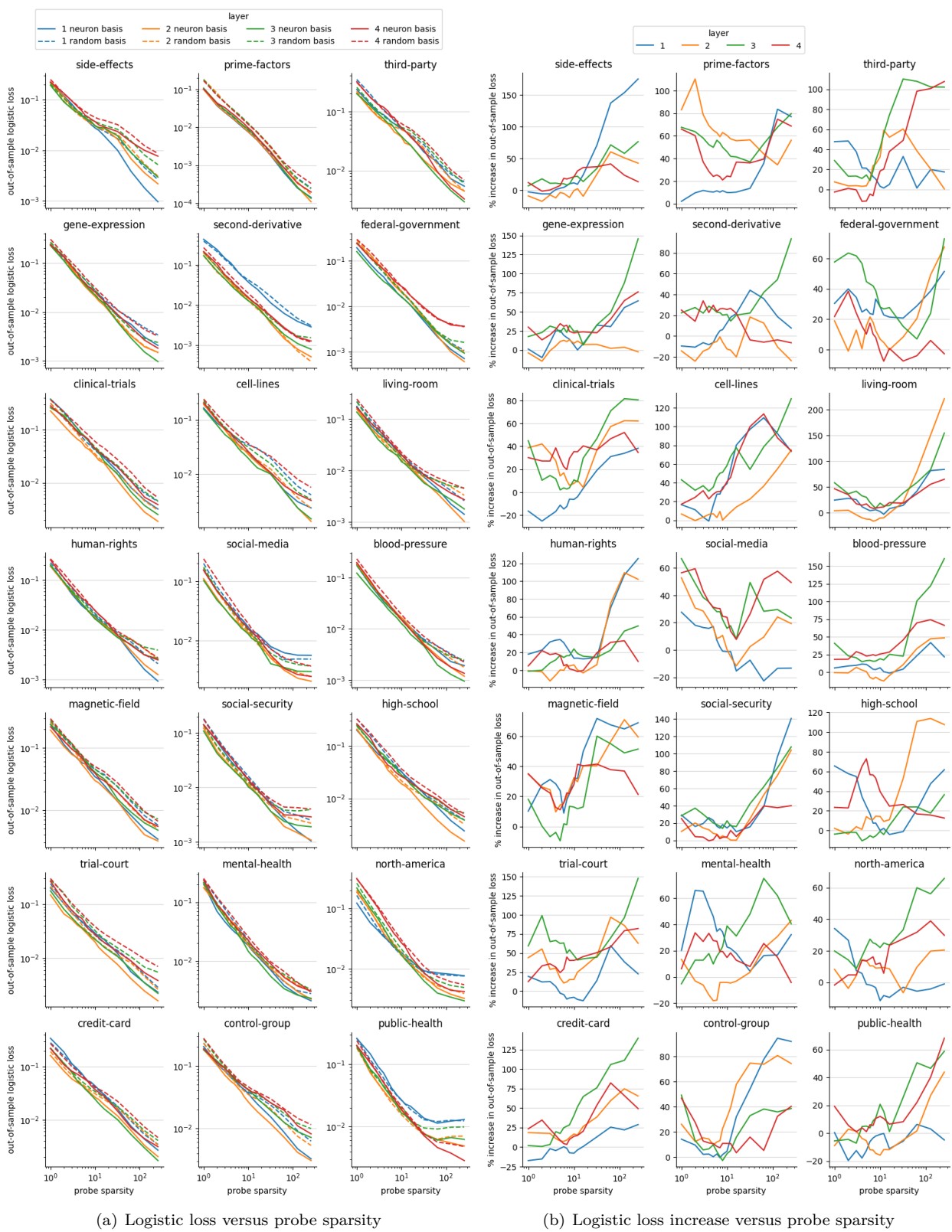

(a) Logistic loss versus probe sparsity

(b) Logistic loss increase versus probe sparsity

Figure 14: Logistic loss for classifying the occurrence of a particular compound word in layers 1-4 of Pythia-1B (in both the neuron basis and the average of random bases) undergo power law scaling until hitting a break point (left). In general, probes in the neuron basis achieve lower overall loss, further suggesting the neuron basis is meaningful (right).

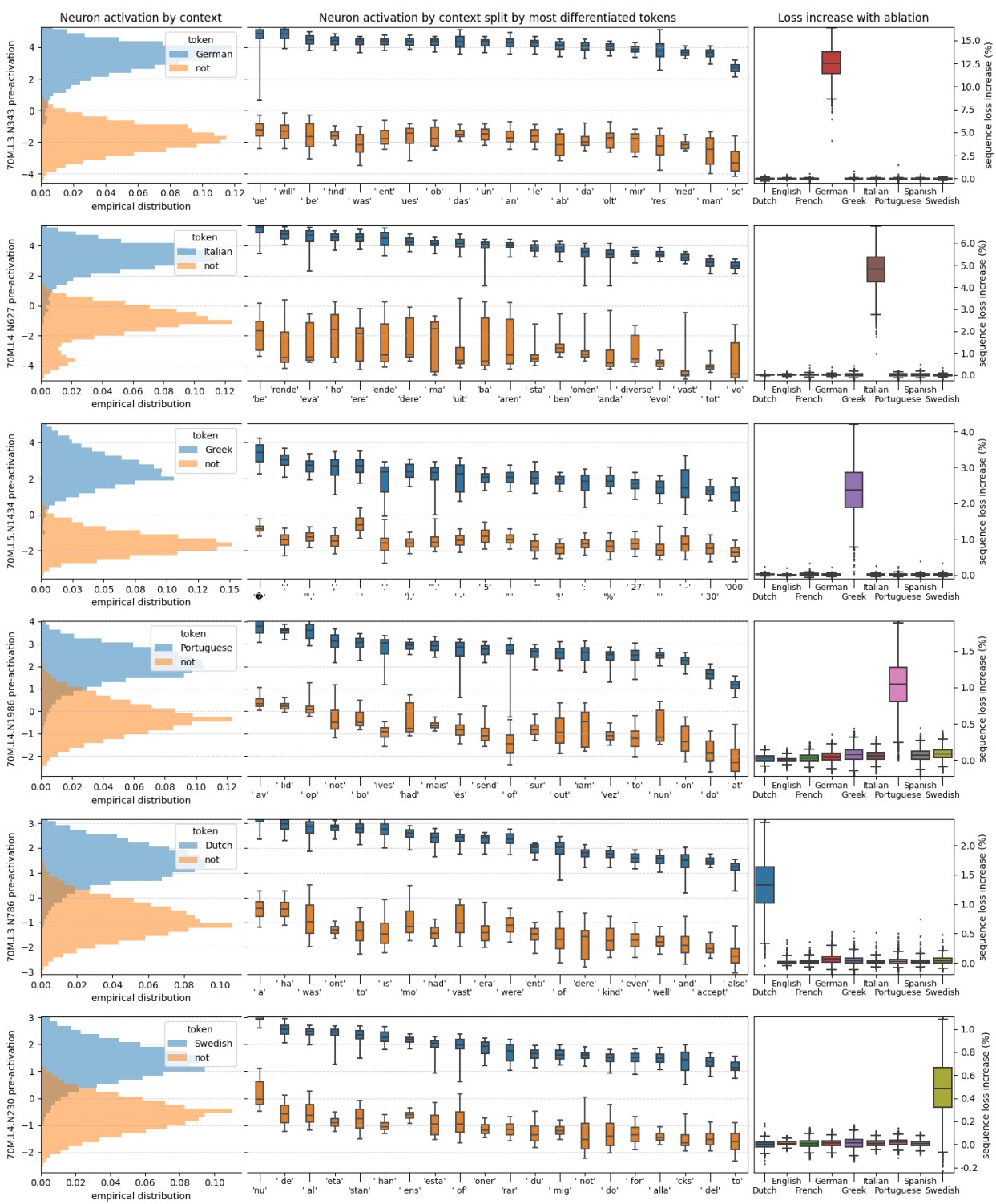

Figure 15: Monosemantic language context neurons in Pythia-70M.

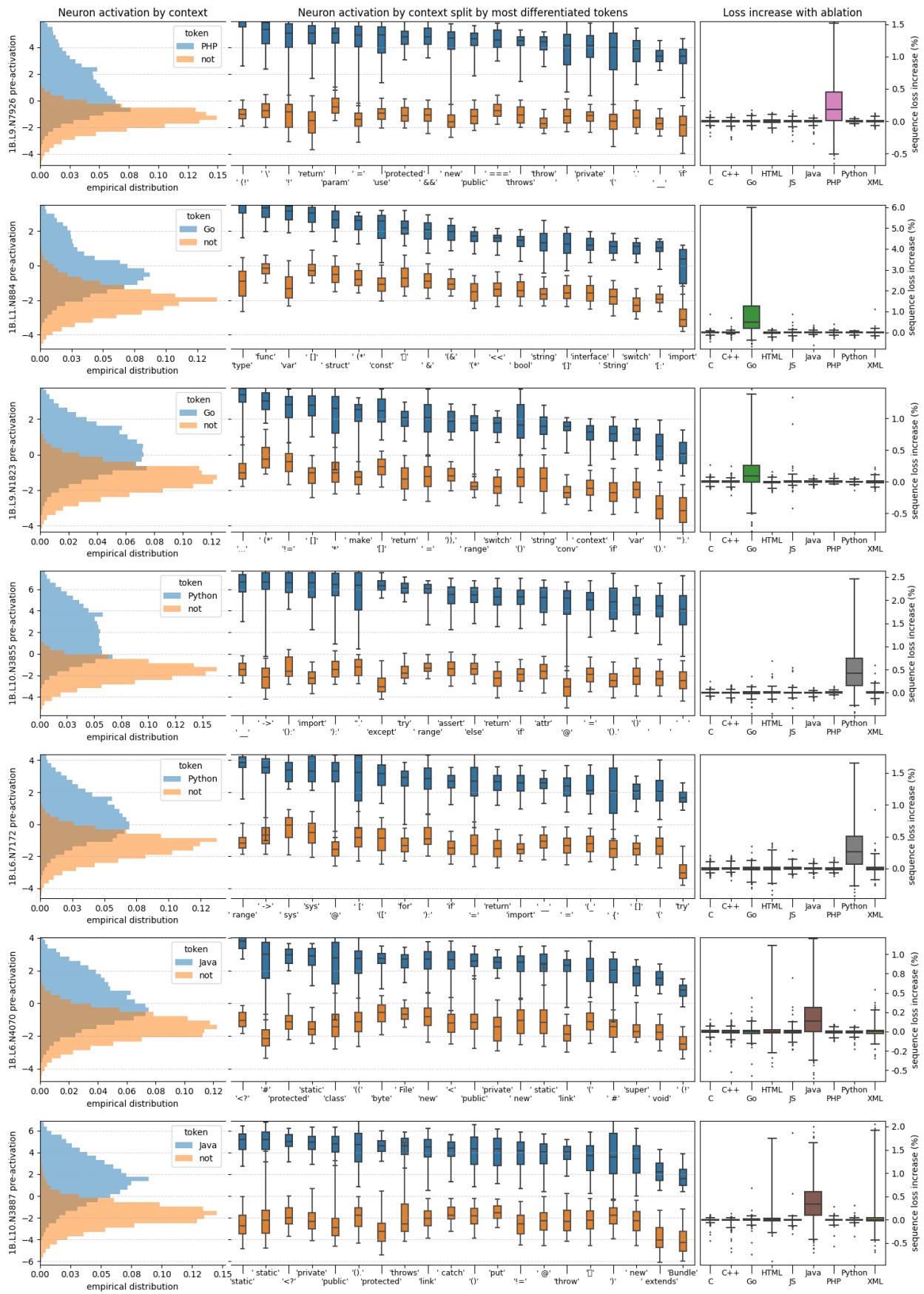

Figure 16: Monosemantic code context neurons in Pythia-1B.

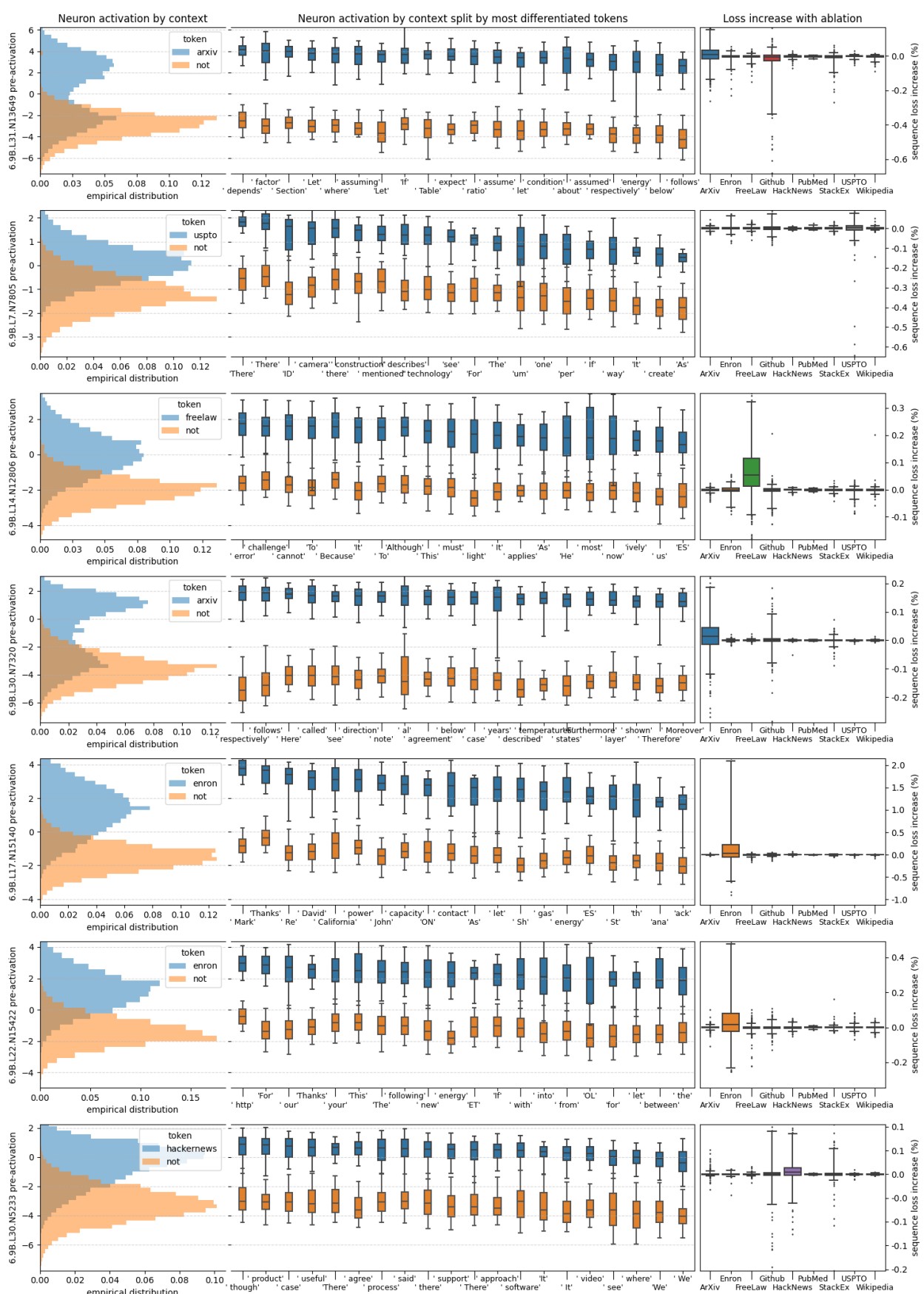

Figure 17: Monosemantic distribution identification context neurons in Pythia-6.9B.

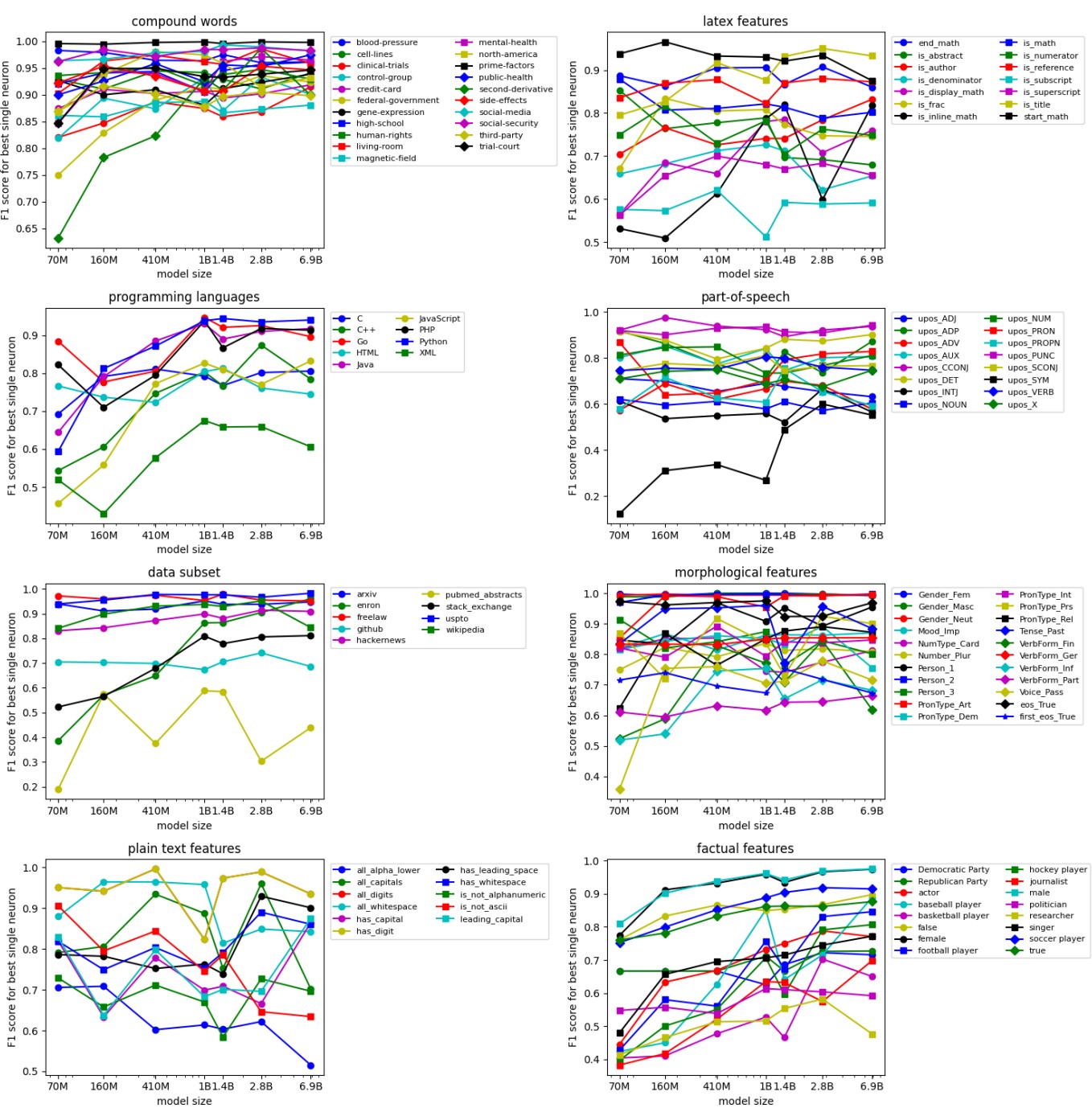

Figure 18: The single neuron with highest F1 classification performance for each feature by model size.

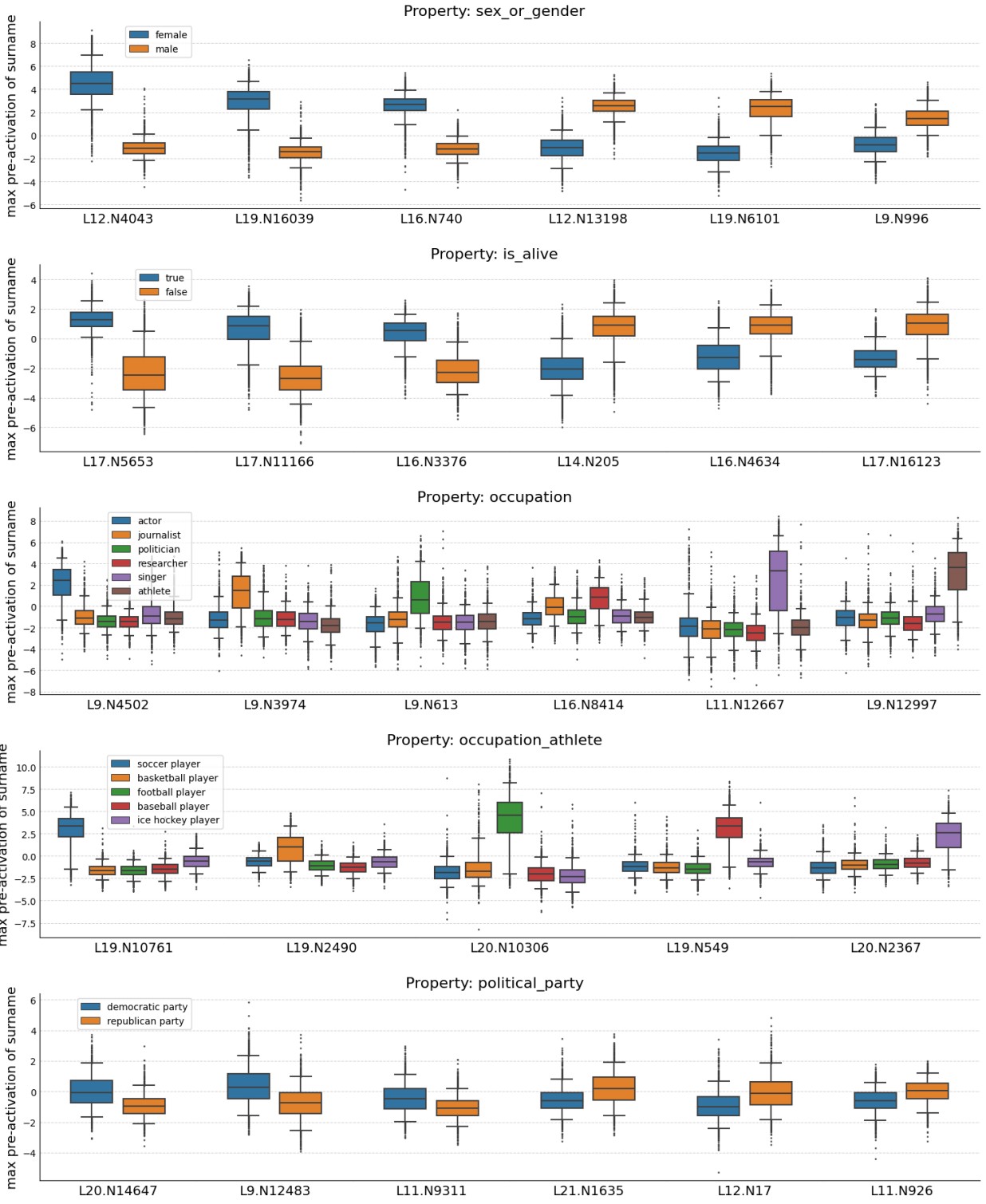

Figure 19: Top neurons in Pythia-6.9B for each category of factual neuron.

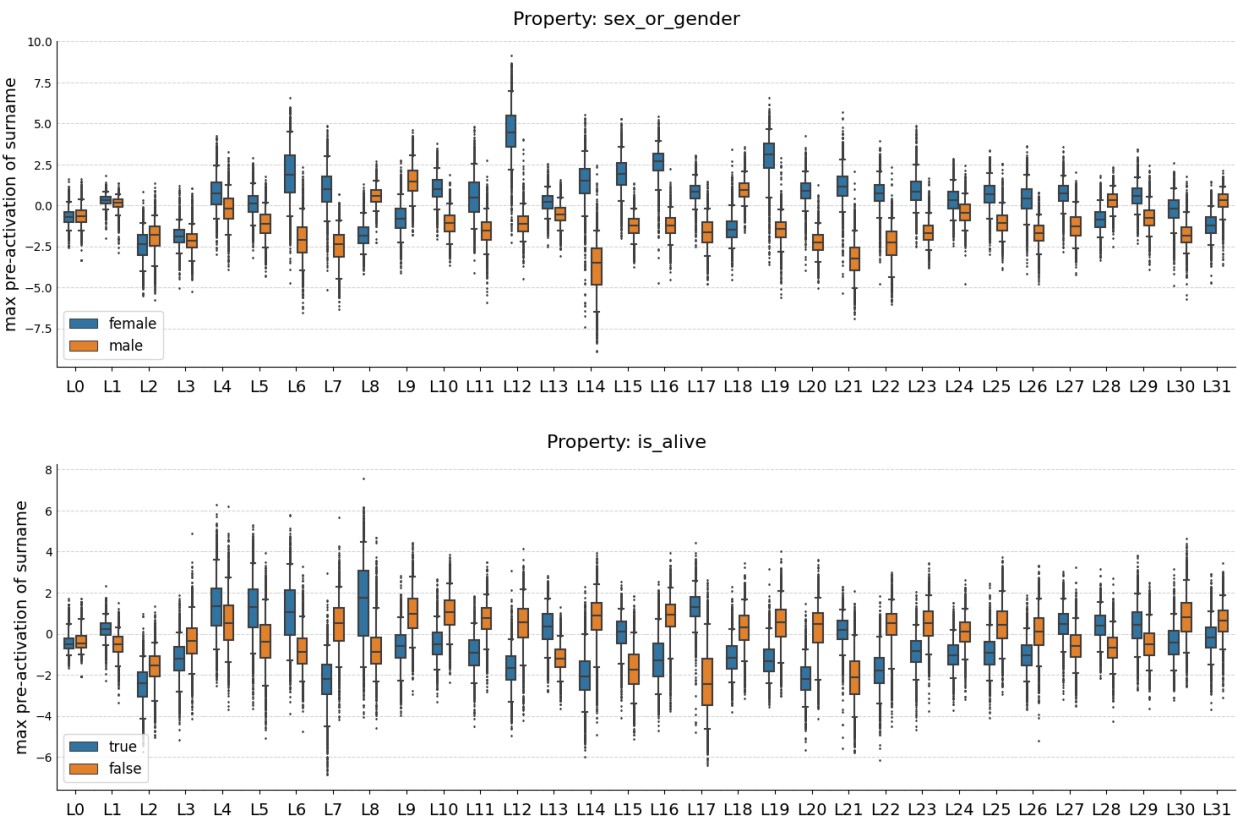

Figure 20: Top neuron for each layer in Pythia-6.9B for the `sex_or_gender` and `is_alive` feature categories.

