# OpenReview forum: "Finding Neurons in a Haystack: Case Studies with Sparse Probing"
_TMLR — Accepted by TMLR_

### Review · Reviewer_JZkJ · 2023-07-05

**Summary Of Contributions:**

This paper studies the problem of interpreting the behaviors of Large Language Models (LLMs) through their internal mechanisms. In particular, the authors use sparse probing to find sets of neurons that can be used to predict particular features (e.g. whether the current token is in French, or if it represents a part of an n-gram). With this toolset, the authors conduct a thorough study of models of varying sizes from the Pythia suite. The authors note multiple interesting phenomena, such as the occurrence of superposition, neurons changing meaning depending on the context and the effects of scale on neuron splitting. The authors conclude their work with a discussion of the results and a careful consideration of the limitations.

**Audience:**

Yes

**Broader Impact Concerns:**

I don't think this paper needs a Broader Impact Statement. There are numerous potential dangers of developing and LLMs, but understanding them does not directly lead to bad outcomes from my perspective.

**Claims And Evidence:**

Yes

**Requested Changes:**

The quality of the paper is high and as of now, I don't have any direct actionable suggestions. I mentioned some of the problems in the "Weaknesses" section above, but I don't think these problems can be solved in the TMLR submission process timeframe.

**Strengths And Weaknesses:**

I find the paper to be a very careful and thorough study of the interpretability of Large Language Models. The proposed investigation techniques are promising and the observations of the authors should be interesting to the community. As such, I think this paper is well suited for this venue.

Strengths:
- The paper aims to tackle the ambitious topic of studying the interpretability of LLMs from the perspective of distributed representations, polysemanticity, and superposition, which were previously well-studied in toy models, but not necessarily in large language models.
- Sparse probing seems to be a promising technique for investigating the interpretability of neurons in LLMs.
- Through their studies, the authors discover multiple novel findings about the nature of neurons in LLMs, including the changes in superposition patterns depending on the layer, the existence of monosemantic neurons, and the relation of the model size and neuron splitting or quantization.
- The authors approach the studied problem rigorously and explicitly list the limitations of their study.
- The paper is very clearly written and is easy to read. In particular, the FAQ section of the Appendix is especially detailed and explains many important concepts and definitions, which is especially important in this very rapidly emerging field.

Weaknesses: \
Note: in general, I find the paper to be quite high quality and the comments below are not necessarily grave errors that should be improved, but rather certain points that I would like to see in e.g. future work.
- Many of the results in this paper are somewhat subjective (how granular should a feature be? How to differentiate between is_french and is_french_noun or is_french_verb?) and as such it is difficult to make rigorous statements. At the same time, the authors discuss these risks and limitations in detail, and I don't see a clear way of bypassing these difficulties.
- The paper focuses only on the interpretability of neurons in the MLPs of LLMs. Studying other aspects, such as the attention mechanism, could provide some additional insights.
- It would be interesting to see if similar conclusions can be drawn for other models than the one from the Pythia suite.

---

> ### Author Response · Authors · 2023-07-19
> **Response to review**
>
> We are glad that the reviewer found our work promising and careful, and thank you for your engagement. Your thoughtful feedback helped us clarify our understanding of what in our results was of interest to the community, and we particularly appreciate your highlighting of our results on neuron splitting and the FAQ.
>
> We agree that the subjectivity of the results is a problem, though consider this a broad issue in the field of interpretability and out of scope of our specific work to address. We think that the attention mechanism is an extremely interesting part of models to investigate, though prior work in mechanistic interpretability has had more success with attention (Elhage et al. 2021; Olsson et al., 2022; Wang et al., 2022a; Nanda et al., 2023; Chughtai et al., 2023) , so we intentionally focused on MLPs. We agree that studying other model suites would be extremely interesting, and intend to open source our code and datasets to enable others to try our techniques on other model suites.

---

### Review · Reviewer_WTD2 · 2023-07-11

**Summary Of Contributions:**

The submission looks at the representations learned in LLMs by using a probing method. They find interesting, if not surprising, features in this representation. They also look at how scale affects their findings.

**Audience:**

Yes

**Broader Impact Concerns:**

Discussed.

**Claims And Evidence:**

Yes

**Requested Changes:**

See above.

**Strengths And Weaknesses:**

** Strengths

* The paper discusses interpretability in the representations of LLM. Two topics that are extremely relevant and important in today's discussions in ML.

* The authors look at a variety of ablations to verify their findings.


** Weaknesses

* Many of the findings are not surprising and as the authors admit follow from simple coding arguments.

* Some statements can use further plots/demonstrations to be convincing. For example, on page 6, the authors claim that "by summing the activations across the three neurons, we achieve nearly perfect separation between the total activation magnitude of the “social security” feature and all other observed token combinations!". However, it is not clear if this perfect separation happens in cases other than "social security" or if all the neurons are required to make the separation.

---

> ### Author Response · Authors · 2023-07-19
> **Response to reviewer WTD2**
>
> We thank you for your comments.
>
> > Many of the findings are not surprising and as the authors admit follow from simple coding arguments.
>
> We agree that our results on superposition for compound word neurons are not surprising from a theoretical level. However, we believe that demonstrating these principles concretely in real-world LLMs provides valuable confirmatory evidence and represents a real research contribution. Our goal is to build up an empirical evidence base for the hypotheses and theoretical principles proposed in prior work. We also note that some findings, like the context neurons, were less anticipated, and demonstrate a mix of compositional and superimposed representations.
>
> > Some statements can use further plots/demonstrations to be convincing
>
> * In figures 9, 10, and 11 we depict 21 other examples of this behavior.
> * In figure 13, we show how well the top 50 individual neurons perform on our bigram classification task, as compared with the top 50 random linear combinations of neurons.
> * Figure 14 shows the probe loss for 21 different bigram classification tasks and different layers as a function of probe sparsity (both in the neuron basis and a random basis)
>
> We also wish to emphasize our discussion in A.10 that it’s unclear if classification performance is the most relevant metric, as we expect the model may want to maximize the separation between classes to minimize interference. This is why in Figure 14 we measured logistic loss rather than classification performance, and observed power law returns to logistic loss as sparsity is increased, until a “kink,” at which point there are no further returns to including more neurons in the classification. We consider this a compelling additional data point beyond just the observation of perfect linear separation.
>
> In light of this comment, we made the pointer to these results clearer in section 5.1.

---

### Review · Reviewer_XC3T · 2023-07-13

**Summary Of Contributions:**

This paper proposed a k-sparse probing method that could locate neurons that are highly relevant for a specific meaning/property of samples (so-called a feature). In this way, potential correlations between the neurons of LLMs and specific features could be found, and the authors perform a number of case studies to illustrate the general properties of large language models (LLMs).

**Audience:**

Yes

**Claims And Evidence:**

No

**Requested Changes:**

1.	Understanding the role of each neuron is not a novel idea. This paper locates highly relevant neurons for a particular feature. It is not novel to propose a method to find the meaning represented by each neuron [1].
2.	More crucially, the proposed explanation conflicts with existing findings in explainable AI.
3.	Many papers argue that neural networks encode features/properties into a few sparse individual neurons. Instead, Kim et al. [2] argued that neural networks used each feature direction for the classification of a certain category. In words, different neurons were densely involved in the classification. A network did not use sparse neurons for the classification. Thus, please conduct experiments to examine whether a network uses sparse neurons for classification. Without the experiments, this study is built upon an unexamined assumption.
4.	I cannot see any solid conclusion or new insights, which guide the learning of DNNs, from the explanation.

[1] David Bau*, Bolei Zhou*, Aditya Khosla, Aude Oliva, and Antonio Torralba. Network Dissection: Quantifying Interpretability of Deep Visual Representations. Computer Vision and Pattern Recognition (CVPR), 2017.

[2] Kim, B., Wattenberg, M., Gilmer, J., Cai, C., Wexler, J., & Viegas, F. (2018, July). Interpretability beyond feature attribution: Quantitative testing with concept activation vectors (TCAV). In International conference on machine learning (pp. 2668-2677). PMLR.

5.	Learning the meaning of each neuron relies on a problematic assumption, i.e., a certain neuron always represents the same feature/property/meaning given different sentences. Given different input sentences, different neurons in the transformer represent different tokens. Therefore, for each neuron in the transformer model, there is not theoretical guarantee to assume that the same neuron always represents the same feature/property/meaning in different sentences.

6.	The definition of “features” should be further clarified. The target feature used to explain the properties of neurons usually represents a too broad concept, e.g., the “is political” feature. Moreover, the **is_political** feature usually depends on the complex patterns of the input. In this way, it is unrealistic to localize certain neurons that correspond to complex patterns of the “is political” feature.

7.	In the first equation in Section 3.1, the structure of the transformer network needs to be represented more clearly. The equation is incorrect. Besides, the “attention” layer, the two “Layer Norm” operations are also ignored in this equation. BTW, why not give each equation an index number? E.g., Equation (1) and Equation (2).

8.	Furthermore, we do not think the authors can faithfully explain the transformer model by ignoring its multi-head attention (MHA) layer.

9.	There are no quantitative metrics and baselines to verify the method. I cannot find any competing methods in experiments. The correctness of the explanation is not evaluated. I do not believe several case studies can justify the correctness of the proposed method.


**Strengths And Weaknesses:**

1. This paper aims to contribute to the general sense LLMs have a tremendous number of correlations between neurons and specific features that humans can understand. The authors propose a k-sparse probing method that could locate neurons related to these interpretable structures.

2. Understanding the role of neurons is an important perspective on the interpretability of neural networks.

---

> ### Author Response · Authors · 2023-07-19
> **Response to reviewer XC3T (General comments on comparison to prior work in vision models)**
>
> Thank you for your comments.
>
> We want to begin by responding to comparisons of our work to prior interpretability and explainability work in vision models (eg, [1], [2]). We think small image classification models and large autoregressive language models have a number of fundamental differences that make comparisons fraught and hence less relevant and potentially even misleading in contextualizing our work. In particular
> * LLMs are much larger– we study models up to 6.9B parameters, with the largest models in existence being in excess of 500B. In contrast, the largest models studied in [2] are only 25M parameters.
> * Different modality –  the pixel values in an image can vary continuously while language is discrete. Further, textual features are sparser and come from a long tail of extremely rare features.
> * Different tasks – [1], [2] refer to vanilla classification models, not autoregressive generative models. Although next token prediction is formulated as a classification problem over a vocabulary of over 50,000 tokens, representations from previous tokens are still used in making predictions for future tokens, and therefore we believe classification is not the right framing for understanding representations.
> * Different architecture components – LLMs have multi-head attention layers, not convolutional kernels, which enables the formation of more sophisticated circuits (Olsson et al., 2022; Wang et al., 2022a)
> * LLMs show emergent behavior (Wei et al., 2022) such as few-shot learning and chain of thought, in a way that was not observed or studied in smaller image models
>
> Given these differences, we would expect differences in the structure of learned representations. Indeed these differences have been observed (Elhage et al. 2022a; Elhage et al. 2022b,) in particular the degree of polysemanticity between language and vision models.

---

> ### Author Response · Authors · 2023-07-19
> **Response to reviewer XC3T (specific comments 1-4)**
>
> Replies to specific comments 1-4
>
> > 1. Understanding the role of each neuron is not a novel idea.
>
> We agree that localizing individual neurons is not a new idea and we cite the past works in this area, focusing on results in LLMs and more recent work in vision models (eg, Bau et al., 2020; Suau et al., 2020; Sajjad et al., 2022; Wang et al., 2022b). We do not see the insight of localizing individual neurons as one of our work's contributions. We further note that per the TMLR guidelines  “novelty of the studied method is not a necessary criteria for acceptance.”
>
> That said, we want to emphasize that part of the purpose of sparse probing was to bridge the gap between individual neuron probing studies and fully dense probing, as a way of studying superposition.
>
> > 2. More crucially, the proposed explanation conflicts with existing findings in explainable AI.
>
> Per the above discussion, we think this conflict was to be expected, and view part of our contribution as surfacing these differences.
>
> > 3. ... A network did not use sparse neurons for the classification. Thus, please conduct experiments to examine whether a network uses sparse neurons for classification. Without the experiments, this study is built upon an unexamined assumption.
>
> The appendix includes a number of experiments across a diverse range of features to examine and verify the sparsity of learned representations.
> * Figure 7 shows the classification performance of sparse probes as a function of sparsity and model size for many different features, as compared to a random k-dimensional projection.
> * Figure 13 shows how well individual neurons perform at our bigram classification task, as compared to “neurons” (ie, basis elements) in a randomly rotated basis.
> * Figure 14 shows the probe loss for different bigram classification tasks and different layers as a function of probe sparsity (both in the neuron basis and a random basis)
>
> Further discussion of these results can be found in A.10 (including weaknesses and confounders) but in summary they show that the neuron basis is meaningful (such that sparsity is plausible in the first place), and that indeed a sparse set of neurons are sufficient for solving our probing classification task.
>
> More broadly, we think that the TCAV results are somewhat narrow in that they focus on features in late layers of classification models, where we would expect to see neural collapse of representations. In early layers of vision models, it has been observed that the feature <=> neuron mapping can be quite sparse and monosemantic (e.g. curve detectors from (Olah et al. 2020)).
>
> As discussed in the introduction, for language models the answer seems to be fairly diverse with a spectrum of representations depending on the nature of the feature (as was predicted by Elhage et al. 2022b). This observation is the motivation for sparse probing, where the analyst can sweep over the sparsity parameter to determine the number of neurons needed to recover the feature. Indeed, we find for contextual features, we find single neurons which are perfectly predictive, whereas for n-grams more neurons are needed.
>
> > 4. I cannot see any solid conclusion or new insights, which guide the learning of DNNs, from the explanation.
>
> The goal of our study was not to improve the capabilities of LLMs – we are motivated by advancing language model interpretability for its own sake, and to better steer and ensure the safety of these models.

---

> ### Author Response · Authors · 2023-07-19
> **Response to reviewer XC3T (specific comments 5-9)**
>
> > 5. Learning the meaning of each neuron relies on a problematic assumption, i.e., a certain neuron always represents the same feature/property/meaning given different sentences.
>
> We do not assume this. One of the main objectives of the paper is to study this very question. Empirically, we find that some neurons do actually correspond to just one feature regardless of context (in particular, the context neurons), but that most neurons are polysemantic, with some being in superposition (eg, the detokenization neurons), with others having multiple facets (eg, the Canada neuron which activates on non-Canadian hockey player names).
>
> We agree the lack of theoretical proof is a limitation, though most interpretability results are empirical and we think that the field lacks a sufficiently formal understanding of interpretability to do a theoretical proof.
>
> > 6. The definition of “features” should be further clarified. The target feature used to explain the properties of neurons usually represents a too broad concept, e.g., the “is political” feature.
>
> We never attempted probing for the is_political feature precisely because we did not think it was possible to operationalize this (as you point out).  Instead we only probed for features where we could provide algorithmic supervision (as in the case of n-grams) or where supervision was provided for us (eg, language in Europarl, part-of-speech from penn treebank, etc). Precise definitions of all features probed for are given in appendix B.2.
>
> It is true that the lack of a fully general definition of a feature is a major challenge for the field, but was not in scope for this work (hence our reliance on feature-specific definitions)
>
> > 7. In the first equation in Section 3.1, the structure of the transformer network needs to be represented more clearly...
>
> Thank you for the correction, we have clarified the equation, and noted the fact we are studying models with parallel attention.
>
> > 8. Furthermore, we do not think the authors can faithfully explain the transformer model by ignoring its multi-head attention (MHA) layer.
>
> We certainly agree that a transformer cannot be fully explained without reference to attention. There has been significant progress with interpretability focused on MHA layers (Elhage et al. 2021, Olsson et al., 2022; Wang et al., 2022a) and a comparative neglect of MLP layers, so we restricted our focus to studying the representations of MLP layers. We think it is out of scope of any individual paper to fully explain a pretrained transformer language model.
>
> > 9. There are no quantitative metrics and baselines to verify the method. I cannot find any competing methods in experiments. The correctness of the explanation is not evaluated. I do not believe several case studies can justify the correctness of the proposed method.
>
> We compare different sparse feature selection methods in Appendix B.5 (Table 4). Given the feature selection methods performed comparably, and our emphasis was on observation, this was moved to the appendix and not emphasized in the main text.
>
> After feature selection, the method is simply a standard logistic regression, for which the “correctness” is simply the classification performance.
>
> For the neurons of our case studies, we perform secondary analyses to verify the interpretations (eg ablations for context neurons, random rotations for n-grams, etc.) Further discussion of validation techniques for interpretation are the subject of Section 5.4.
>
> We do agree that these analyses are in general ad hoc, and therefore limit their applicability (discussed in 6.1 and 6.2), but nonetheless provide evidence for our interpretations.

---

### Author Response · Authors · 2023-08-29
**Gentle reminder to respond**

We would like to kindly remind the reviewers to respond to our rebuttals before the end of the review period. Thank you.

---

### Decision · Action_Editors · 2023-10-13

**Recommendation:** Accept with minor revision

**Comment:**

Among the three reviewers, two reviewers are positive. Reviewer JZkJ thinks highly of this paper, noting the meticulous study and careful and extensive discussion. Reviewer XC3T remains concerned with novelty, conclusion and the method. However, I think the paper is novel by applying sparse probing to LLMs. Linear probing is a valid method for understanding LLMs. It is possible that individual neurons may predict features. Therefore, I would like to override Reviewer XC3T's recommendation.

Overall I think the paper makes a useful contribution to a timely topic, and I recommend acceptance. But I do hope that the authors will revise their paper to address the concerns of Reviewer XC3T.

**Audience:**

This paper studies the interpretability of large language models, and it should be of interest to the TMLR's audience in general.

**Claims And Evidence:**

This paper studies the interpretation of large language models using sparse probing. The authors find neurons that can predict certain features. The authors observe some interesting phenomena in large language models in their experiments. The experiments are thorough and the claims are well supported by evidence.